# Vevo: Controllable Zero-Shot Voice Imitation with Self-Supervised Disentanglement

Xueyao Zhang[1][*]  Xiaohui Zhang[2]  Kainan Peng[2]  Zhenyu Tang[2]  Vimal Manohar[2],
Yingru Liu[2]  Jeff Hwang[2]  Dangna Li[2]  Yuhao Wang[2]  Julian Chan[2]  Yuan Huang[2]
Zhizheng Wu[1][†]  Mingbo Ma[2]
[1]The Chinese University of Hong Kong, Shenzhen    [2]Meta AI

## Abstract

The imitation of voice, targeted on specific speech attributes such as timbre and speaking style, is crucial in speech generation. However, existing methods rely heavily on annotated data, and struggle with effectively disentangling timbre and style, leading to challenges in achieving controllable generation, especially in zero-shot scenarios. To address these issues, we propose Vevo, a versatile zero-shot voice imitation framework with controllable timbre and style. Vevo operates in two core stages: (1) *Content-Style Modeling*: Given either text or speech's *content* tokens as input, we utilize an autoregressive transformer to generate the *content-style* tokens, which is prompted by a style reference; (2) *Acoustic Modeling*: Given the *content-style* tokens as input, we employ a flow-matching transformer to produce acoustic representations, which is prompted by a timbre reference. To obtain the content and content-style tokens of speech, we design a fully self-supervised approach that progressively decouples the timbre, style, and linguistic content of speech. Specifically, we adopt VQ-VAE [1] as the tokenizer for the continuous hidden features of HuBERT [2]. We treat the vocabulary size of the VQ-VAE codebook as the information bottleneck, and adjust it carefully to obtain the disentangled speech representations. Solely self-supervised trained on 60K hours of audiobook speech data, without any fine-tuning on style-specific corpora, Vevo matches or surpasses existing methods in accent and emotion conversion tasks. Additionally, Vevo's effectiveness in zero-shot voice conversion and text-to-speech tasks further demonstrates its strong generalization and versatility.

## 1 Introduction

The imitation of voice has long been an important issue in the field of speech generation. This includes the imitation of speaker identity [3, 4], the imitation of speaking style such as accent [5, 6] or emotion [7], and a broader concept of voice cloning such as in zero-shot text-to-speech (TTS) task [8]. These techniques have a wide range of applications, including spoken language learning [5, 6, 9], voice anonymization [10], voice assistants [11, 12], and video dubbing [11, 12, 13].

To achieve targeted and controllable imitation over various speech attributes, many studies focuses on factorizing speech into multiple sub-spaces [14, 15, 16, 17]. In this work, we follow this idea and decompose speech into three key attributes: linguistic content (*what to speak*), style (*how to speak*), and timbre (*who speaks*). Based on this, we define three zero-shot speech generation tasks (Table 1): (1) **Timbre Imitation**: Given a speech as source, imitate only the timbre of the reference speech while preserving the linguistic content and speaking style. It can be adopted in voice conversion that only spectral aspects of speech are converted [3]. (2) **Style Imitation**: Given a speech as source, imitate only the speaking style of the reference speech while preserving the content and the timbre. It can be adopted in accent conversion [5] and emotion conversion [7]. (3) **Voice Imitation**: Given either a speech (i.e., *conversion task*) or text (i.e., *synthesis task*) as source, imitate both the timbre and style of the reference speech while preserving the content. It can be adopted in voice conversion that both spectral and prosodic aspects of speech are converted [3, 4] and zero-shot TTS [8].

---

*[*]Work accomplished during the internship at Meta.
[†]Corresponding author.

Table 1: Definitions of zero-shot timbre, style, and voice imitation tasks.

| Task | Source ($i$) | Reference ($r$) | Attribute(s) to Imitate | Target | Related Areas |
|---|---|---|---|---|---|
| Timbre Imitation | | | Timbre | $\mathcal{W}(c_i, s_i, t_r)$ | Voice Conversion |
| Style Imitation | $\mathcal{W}(c_i, s_i, t_i)$ | $\mathcal{W}(c_r, s_r, t_r)$ | Style | $\mathcal{W}(c_i, s_r, t_i)$ | Accent Conversion, Emotion Conversion |
| Voice Imitation | | | Timbre and Style | $\mathcal{W}(c_i, s_r, t_r)$ | Voice Conversion |
| | $\mathcal{T}(c_i)$ | | | | Text to Speech |

> \* $\mathcal{W}$ and $\mathcal{T}$ denote speech and text. $c_i$, $s_i$, and $t_i$ represent the linguistic content, style, and timbre of the source $i$. Similarly, $c_r$, $s_r$, and $t_r$ represent the linguistic content, style, and timbre of the reference $r$.

To address these imitation tasks, existing work has explored approaches including learning the conversion between parallel corpus [9, 18, 19, 20, 21], disentangled representation learning [2, 14, 17, 22, 23, 24], and large-scale in-context learning [11, 25, 26, 27, 28]. However, these approaches still suffer from the following limitations. Firstly, for the style imitation, existing methods rely heavily on supervision with annotated data, which is hard to collect and scale up. This reliance includes the use of parallel corpus [9, 20, 21], style labels (such as categories of accent [20, 21, 29] or emotion [30, 31]), and textual transcriptions [29, 30, 31, 32]. Moreover, achieving *zero-shot* style imitation—where a system can imitate an accent, emotion, or other speaking styles from just a few seconds of speech—remains a significant challenge. Secondly, the decoupling of timbre and style in existing methods is still insufficient, making it challenging to control them independently, unless mitigated by some timbre (or style) perturbations or additional fine-tuning stages [11, 13, 33].

Motivated by the above, this paper proposes Vevo, a versatile zero-shot voice imitation framework with controllable timbre and style (Figure 1). It can serve as a unified framework for a wide range of zero-shot speech generation tasks. Vevo consists of two core stages: (1) **Content-Style Modeling** (*Content to Content-Style*): Given a speech prompt as style reference, we generate *content-style* tokens from the input *content* tokens (or the input text). We employ the decoder-only autoregressive transformer [34, 35], leveraging its powerful capability of continued generation to model style. (2) **Acoustic Modeling** (*Content-Style to Acoustic*): Given a speech prompt as timbre reference, we generate acoustic representations (such as Mel spectrograms) from the input of *content-style* tokens. We use a flow-matching transformer [36, 37], which has been verified to excel in in-context learning and reconstructing high-quality audio [12, 24, 27, 38], to achieve timbre-controllable generation.

To obtain the *content* and *content-style* tokens of speech, we design a self-supervised method to decouple the timbre, style, and linguistic content gradually, which is similar to a progressive information filtering: (1) We firstly investigate the commonly used self-supervised speech pre-trained model, HuBERT [2]. We find that its **continuous** hidden features contain rich information about timbre, style, and linguistic content (Section 4.1), making it a suitable initial stage for information filtering. (2) Inspired by existing works for disentangling speaker-agnostic representations [1, 17, 39, 40], we employ VQ-VAE [1] as a tokenizer for HuBERT to filter out timbre, resulting in **content-style tokens**. (3) Furthermore, we propose that the vocabulary size of the VQ-VAE codebook can function as the "width" of the information bottleneck [22]. By reducing the vocabulary size, we can narrow the bottleneck and filter out not only timbre but also significant style information, thereby obtaining **content tokens**. Besides, we propose to reduce the consecutive duplicate units [41] of the content tokens, called *duration reduction*, to further remove some style patterns such as unit-level duration.

The contributions of this paper are summarized as follows:

- We introduce a fully self-supervised approach that progressively decouple timbre, style, and linguistic content of speech. The resulting content-style tokens and content tokens enhance controllability in downstream speech generation tasks, particularly for timbre and style.

- We propose Vevo, a unified framework that enables versatile, controllable zero-shot voice imitation tasks. It significantly reduces the reliance on annotated corpora, facilitating self-supervised training and in-context learning that can easily be scaled up.

- Pre-trained on 60K hours of audiobook speech data without any fine-tuning on style-specific corpora, Vevo matches or even surpasses existing methods in accent and emotion conversion tasks – notably, through zero-shot imitation. Additionally, Vevo's effectiveness in voice conversion and text-to-speech tasks further demonstrates its strong generalization and versatility.

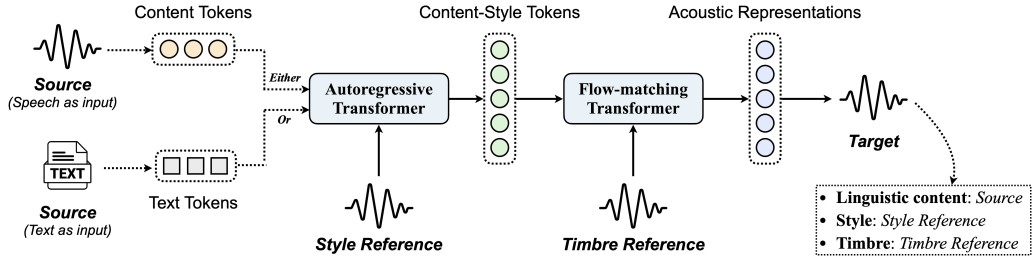

Figure 1: Vevo inference pipeline. Notably, it can take *either* speech *or* text as input, and perform zero-shot imitation with controllable linguistic content (controlled by the source), style (controlled by the style reference), and timbre (controlled by the timbre reference) in a single forward pass.

## 2 RELATED WORK

**Controllable Voice Imitation**   We focus primarily on how existing works approach the imitation of two key speech attributes: timbre and style. (1) **Imitation of Timbre**: As a crucial aspect of speaker identity, timbre imitation has been extensively explored within the voice conversion (VC) field. Most studies aim to utilize the speaker-agnostic representations such as PPG features [20, 42] or some self-supervised representations [43, 44], and use models including GAN [45, 46], auto-encoder [14, 22], and diffusion models [47, 48] to achieve timbre imitation. (2) **Imitation of Style**: In terms of style imitation, accent and emotion are two widely studied attributes. For conversion tasks (with speech as input), classic approaches often involve learning the conversion between parallel corpus [9, 19, 20, 21]. Additionally, many studies aim to obtain the style-agnostic features, such as pushing them to be close to textual transcriptions [30, 31, 32, 49]. Besides, leveraging automatic speech recognition (ASR) models can transform conversion tasks into synthesis tasks, allowing the injection of style label's embeddings into TTS models to achieve style imitation [29, 50]. In conclusion, these existing approaches often rely on annotated data and struggle to achieve *zero-shot* style imitation. (3) **Imitation of both Timbre and Style**: In VC, some works suggest adopting a sequence-to-sequence formulation [51, 52] or introducing an additional modeling for prosody features [48, 53] to achieve both timbre and style imitation. However, these models still have significant room for improvement in both quality and style imitation. Recent advances in zero-shot TTS have greatly improved voice imitation and cloning. They leverage large-scale in-context learning to mimic all speech attributes of a reference prompt, including timbre and style, with high quality and speaker similarity [11, 13, 16, 17, 26, 33]. Nonetheless, it is challenging to obtain the speech representations disentangled timbre and style effectively [23, 33], leading to inadequate targeted control of these attributes. For instance, using the existing representations directly for VC tasks will lead to timbre leakage, unless mitigated by timbre perturbation or an additional fine-tuning stage [11, 13].

**Disentangled Speech Representation**   There are many studies aim to decouple linguistic content, timbre, and style. Existing work on obtaining disentangled speech representations can generally be categorized into several approaches: (1) Knowledge distillation using auxiliary tasks such as ASR, F0 prediction, and speaker verification [15, 17, 23], (2) Model architecture design based on information bottlenecks, including careful adjustments to hidden layer dimensions [14, 22] or vector quantization methods like K-means [2, 54, 55] or VQ-VAE [1, 15, 17, 39, 40], and (3) Perturbation of acoustic signals [56, 57, 58]. Besides, existing works also leverage additional learning strategies including adversarial learning [17, 23], comparative learning [23, 59], and mutual information minimization [40, 60, 61] to enhance disentanglement effectiveness. However, existing work still has two main weaknesses. On one hand, as mentioned earlier, finding suitable representations for downstream generation tasks that can effectively decouple timbre and style remains quite challenging. On the other hand, how to design voice imitation models that can control specific attributes based on these disentangled speech representations has been scarcely explored.

## 3 METHODOLOGY

### 3.1 VQ-VAE TOKENIZER FOR HUBERT

**Motivation**   To disentangle representations of different speech attributes, we adopt a VQ-VAE tokenizer [1] due to its demonstrated potential in disentangling high-level information within speech

such as speaker-invariant features [1, 17, 39]. In speech domain, it is common practice to apply VQ-VAE either directly on the raw waveform [1, 17, 39] or on the self-supervised learning (SSL) based speech representations [12, 15, 62]. In this work, we choose to apply VQ-VAE based on SSL representations – specifically, HuBERT [2]. The reasons are two fold: (1) HuBERT's continuous hidden features already contain rich information about timbre, style, and linguistic content, making them well-suited for reconstructing acoustic representations such as Mel spectrograms (Section 4.1); (2) Self-supervised learning on speech could be also treated as a high-level knowledge distillation. VQ-VAE enables us to further information filtering and disentangling for the SSL features.

**Architecture**  The VQ-VAE consists of three components: Encoder, Vector Quantization (VQ), and Decoder. Formally, given the codebook $\boldsymbol{E} = [\boldsymbol{e}_1, \boldsymbol{e}_2, \ldots, \boldsymbol{e}_K]$ whose vocabulary size is $K$, taking HuBERT hidden features $\boldsymbol{x}$ as input, we get the reconstructed $\hat{\boldsymbol{x}}$ after the three modules:

$$
\begin{aligned}
\boldsymbol{z}_e(\boldsymbol{x}) &= \text{Encoder}(\boldsymbol{x}), \\
\boldsymbol{z}_q(\boldsymbol{x}) &= \boldsymbol{e}_k, \text{ where } k = \arg\min_j \|\boldsymbol{z}_e(\boldsymbol{x}) - \boldsymbol{e}_j\|_2, \\
\hat{\boldsymbol{x}} &= \text{Decoder}(\boldsymbol{z}_q(\boldsymbol{x})),
\end{aligned}
\tag{1}
$$

where $\boldsymbol{z}_q(\boldsymbol{x})$ is the quantized representation (i.e., token) of $\boldsymbol{z}_e(\boldsymbol{x})$ after VQ. The loss function consists of the reconstruction loss (whose weight is $\lambda$) and quantization loss (whose weight is $\beta$):

$$
\mathcal{L} = \lambda\|\boldsymbol{x} - \hat{\boldsymbol{x}}\|_2^2 + \beta\|\boldsymbol{z}_e(\boldsymbol{x}) - \boldsymbol{z}_q(\boldsymbol{x})\|_2^2.
\tag{2}
$$

Note that there is no real gradient defined for $\boldsymbol{z}_q(\boldsymbol{x})$. We could utilize the straight-through gradient estimator or exponential moving average (EMA) as the optimization algorithm [1]. In this paper, we follow the design in [62, 63] and use the EMA algorithm. We describe the specific module design of VQ-VAE in Appendix B.1. Notably, the VQ-VAE model does not contain any downsampling or upsampling operations, thus preserving the sequence length of the input $\boldsymbol{x}$. In other words, for the 50 Hz frame-level HuBERT features [2], we can also get 50 Hz frame-level tokens after VQ.

**Analysis of the Vocabulary Size of Codebook**  The quantization of HuBERT hidden features by VQ-VAE can be viewed as a form of *lossy compression*. Inspired by AutoVC [22], we propose that the vocabulary size of the VQ codebook acts as an information bottleneck. If the input $\boldsymbol{x}$ possesses sufficient speech information, reducing the vocabulary size $K$ from infinity to zero: (1) **When $K \to \infty$**, we consider the bottleneck to be extremely wide, capable of accommodating all information without any loss. (2) **As $K$ decreases**, more low-level acoustic information begins to be lost, such as spectral features related to timbre or prosodic features related to style. At a certain reduced $K$, only the highest-level, most abstract information like linguistic content is preserved within $\boldsymbol{x}$. (3) **When $K \to 0$**, the bottleneck becomes exceedingly narrow, filtering out even high-level information like linguistic content. We validate the above hypothesis through experiments on the zero-shot timbre imitation task (Section 4.1). Interestingly, as we progressively reduce $K$, we observe that timbre information is the first to be filtered out (assuming when $K = K_s$), from which we derive the *content-style* tokens. Subsequently, most style information is filtered, and ultimately, almost only the highest-level linguistic content information is retained (assuming when $K = K_c$), from which we derive the *content* tokens. We refer to the VQ-VAE model whose $K = K_s$ as the content-style tokenizer $\boldsymbol{Q}_s$, and the model whose $K = K_c$ as the content tokenizer $\boldsymbol{Q}_c$.

## 3.2 CONTENT-STYLE MODELING (CONTENT TO CONTENT-STYLE)

During the content-style modeling stage, our goal is to transform the content token of speech (or text) into content-style tokens, which is prompted by a style reference. This can be formulated as a sequence-to-sequence generation task. For this stage, we employ a decoder-only autoregressive (AR) transformer, known for its powerful capability in such tasks [11, 34, 35]. In this section, we will focus only on cases where speech's content tokens are used as input (Figure 2). The scenarios where text serves as input will be discussed in Appendix B.3.

**Duration Reduction**  Given a speech input $u$, we denote the content and content-style tokens as $\boldsymbol{Q}_c(u)$ and $\boldsymbol{Q}_s(u)$. Both of them are 50 Hz frame-level representations of equal length. In the content-style modeling stage, $\boldsymbol{Q}_s(u)$ is used as the output. However, instead of using $\boldsymbol{Q}_c(u)$, we apply a *Duration Reduction* strategy to it, yielding the reduced $\boldsymbol{Q}'_c(u)$ as the input. Specifically, we merge the consecutive duplicate units of $\boldsymbol{Q}_c(u)$ into one. For instance, if $\boldsymbol{Q}_c(u) = [\boldsymbol{e}_1, \boldsymbol{e}_1, \boldsymbol{e}_1, \boldsymbol{e}_2, \boldsymbol{e}_3, \boldsymbol{e}_3]$, it will be condensed to $\boldsymbol{Q}'_c(u) = [\boldsymbol{e}_1, \boldsymbol{e}_2, \boldsymbol{e}_3]$. This strategy offers significant

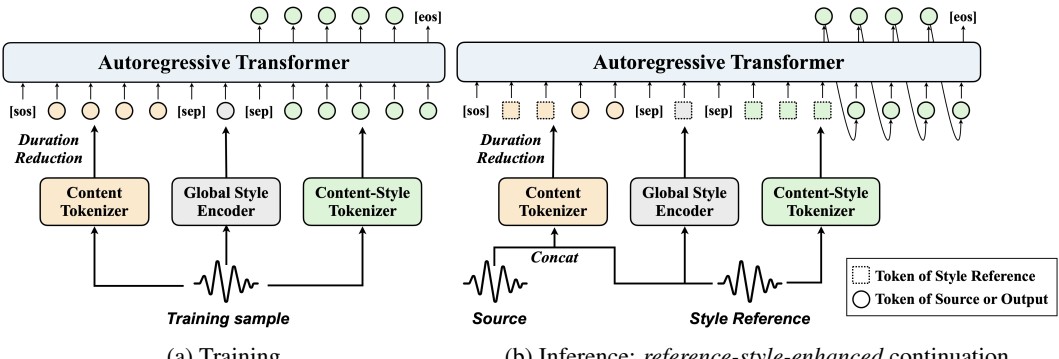

Figure 2: Content-style modeling based on autoregressive transformer. During inference, we employ both global style encoder and content-style tokenizer to enhance the effect of the style reference.

benefits: (1) It further filters out style-specific information within $\boldsymbol{Q}_c(u)$ such as the unit-level duration. Some studies also point out that such a reduction could aid in reducing accents and other style elements [41]; (2) It resolves the model's challenge with learning changes in sequence length before and after style modeling when $\boldsymbol{Q}_c(u)$ and $\boldsymbol{Q}_s(u)$ are always equal in length; (3) It shortens the overall sequence length, which is beneficial to model context for transformer.

**Global Style Encoder**    We design a global style encoder to capture the global style guidance from the speech input $u$, producing a style embedding (denoted as $\boldsymbol{g}(u)$). Its advantage comes from the flexibility during inference: if we aim to optimize inference speed and reduce memory usage, we can rely solely on this style embedding for style guidance, named as *reference-global-guided* continuation (Figure 5). However, to maximize the performance of style imitation, in addition to using $\boldsymbol{g}(u)$, we can also append the style reference's content-style tokens into the input sequence to enhance its effect, named as *reference-style-enhanced* continuation (Figure 2b). The global style encoder consists of WavLM-based representation layers and TDNN-based feature extraction layers [64, 65]. We describe the detailed module design in Appendix B.2.

**Training and Inference**    During training, we conduct self-supervised learning on speech data. The input sequence of transformer is $[\langle\text{SOS}\rangle, \boldsymbol{Q}_c'(u), \langle\text{SEP}\rangle, \boldsymbol{g}(u), \langle\text{SEP}\rangle, \boldsymbol{Q}_s(u)]$. We only perform the next token prediction on the last $[\langle\text{SEP}\rangle, \boldsymbol{Q}_s(u)]$, with the ground truth being $[\boldsymbol{Q}_s(u), \langle\text{EOS}\rangle]$. Here, $\langle\text{SOS}\rangle$, $\langle\text{SEP}\rangle$, and $\langle\text{EOS}\rangle$ are treated as three special tokens in language model [66]. During inference, for a source speech $u_i$ and a style reference $u_{sr}$, we can conduct the reference-style-enhanced continuation (Figure 2b) by feeding the input sequence $[\langle\text{SOS}\rangle, \boldsymbol{Q}_c'(u_{sr} \oplus u_i), \boldsymbol{g}(u_{sr}), \boldsymbol{Q}_s(u_{sr})]$ for autoregressive generation, where $\oplus$ means the concatenation. For reference-global-guided continuation (Figure 5), the input sequence becomes $[\langle\text{SOS}\rangle, \boldsymbol{Q}_c'(u_i), \boldsymbol{g}(u_{sr})]$.

### 3.3 ACOUSTIC MODELING (CONTENT-STYLE TO ACOUSTIC)

During the acoustic modeling stage, prompted by a timbre reference, we aim to transform the content-style tokens to Mel spectrograms. We adopt a flow matching transformer [34, 35, 36] (Figure 3), which has been verified to be effective in in-context learning and reconstructing high-quality acoustic representations [12, 24, 27, 38].

During training, given a speech $u$ and its Mel spectrogram $\boldsymbol{y}_1$, we randomly select a part of $\boldsymbol{y}_1$ as the timbre reference (denoted as $\boldsymbol{y}_1^{ctx}$), and aim to reconstruct the other part (denoted as $\boldsymbol{y}_1^{mis}$) conditioned on $\boldsymbol{y}_1^{ctx}$ and the content-style tokens $\boldsymbol{Q}_s(u)$. In other words, we aim to model the conditional probability $p(\boldsymbol{y}_1^{mis}|\boldsymbol{y}_1^{ctx}, \boldsymbol{Q}_s(u))$. Specifically, we follow Voicebox [27] and use a temporal span masking strategy: $\boldsymbol{y}_1^{mis} = \boldsymbol{m} \odot \boldsymbol{y}_1$, and $\boldsymbol{y}_1^{ctx} = (\boldsymbol{1} - \boldsymbol{m}) \odot \boldsymbol{y}_1$, where $\boldsymbol{m}$ is a binary temporal mask that is of the same length as $\boldsymbol{y}_1$, and $\odot$ means the element-wise multiplying operation. During inference, given a source speech $u_i$ and a timbre reference $u_{tr}$, all the source's Mel spectrogram will be masked (i.e., $\boldsymbol{y}_1^{mis}$). The input conditions become the timbre reference's Mel spectrogram (i.e., $\boldsymbol{y}_1^{ctx}$) and the concatenated content-style tokens $\boldsymbol{Q}_s(u_i \oplus u_{tr})$. This enables the generated target to preserve the linguistic content and style of $u_i$, and the timbre of $u_{tr}$ (Figure 3b).

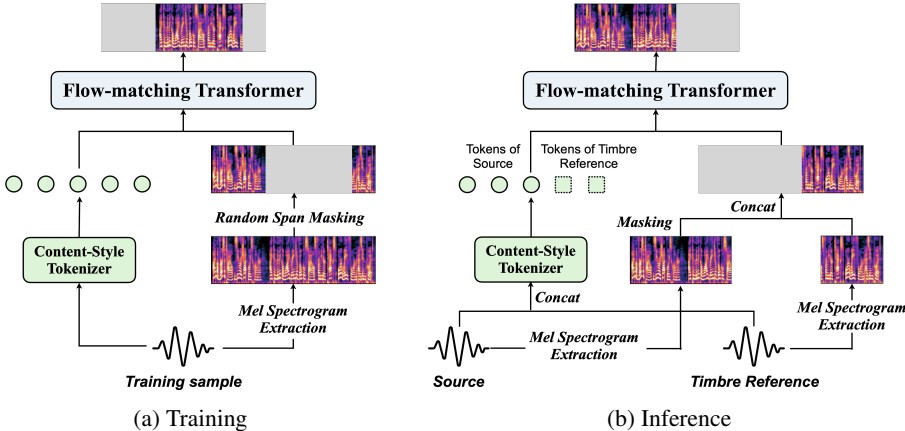

(a) Training  (b) Inference

Figure 3: Acoustic modeling based on a flow-matching transformer. During inference, we append the timbre reference to the rightmost (or leftmost) end, enabling timbre-controllable generation.

We use the conditional flow matching algorithms based on optimal transport path, which is widely adopted in related works [12, 24, 27]. The loss function is defined as:

$$\mathcal{L}_{cfm} = \mathbb{E}_{t,\boldsymbol{m},\boldsymbol{y}_0,\boldsymbol{y}_1} \left\| \frac{d\boldsymbol{y}_t}{dt} - f_t(\boldsymbol{y}_t, t, \boldsymbol{y}_1^{ctx}, \boldsymbol{Q}_s(u)) \right\|_2^2, \tag{3}$$

$$\text{where } \boldsymbol{y}_t = (1 - (1-\sigma)t) \cdot \boldsymbol{y}_0 + t \cdot \boldsymbol{y}_1,$$

where $t$ is the time step that is sampled from the uniform distribution $\mathcal{U}(0,1)$, $\boldsymbol{y}_0$ is a noise sampled from standard Gaussian distribution, $f_t(\cdot)$ is the vector filed (which is estimated by transformer). and $\sigma$ is a small constant of the optimal transport (OT) path. Notably, the frame rates of the content-style tokens $\boldsymbol{Q}_s(u)$ and the Mel spectrogram $\boldsymbol{y}_1$ could be different. We follow [44] and use a simple signal resampling operation to align them. Then we use the adding operation to fuse their frame-level features. We describe the detailed module design in Appendix B.4. After obtaining the Mel spectrogram, we utilize a BigVGAN [67] vocoder to produce the waveform (Appendix B.5).

## 3.4 Vevo for Various Zero-Shot Imitation Tasks

Assume that during the content-style modeling and acoustic modeling stages, we have obtained pre-trained models $\mathcal{M}_{style}$ and $\mathcal{M}_{acoustic}$ respectively. We can then adjust only the inference pipeline to apply Vevo to various zero-shot imitation tasks. Given the source speech $u_{\boldsymbol{i}}$ (or text $\mathcal{T}_{\boldsymbol{i}}$) and the reference $u_{\boldsymbol{r}}$, we can utilize the following variants of Vevo to achieve zero-shot timbre, style, and voice imitation tasks (" $\xrightarrow{u} \mathcal{M}$ " means that the model $\mathcal{M}$ is prompted by $u$ to generate):

- **Vevo-Timbre** for timbre imitation: $\boldsymbol{Q}_s(u_{\boldsymbol{i}}) \xrightarrow{u_{\boldsymbol{r}}} \mathcal{M}_{acoustic}$
- **Vevo-Style** for style Imitation: $\boldsymbol{Q}'_c(u_{\boldsymbol{i}}) \xrightarrow{u_{\boldsymbol{r}}} \mathcal{M}_{style} \xrightarrow{u_{\boldsymbol{i}}} \mathcal{M}_{acoustic}$
- **Vevo-Voice** for voice imitation (conversion task): $\boldsymbol{Q}'_c(u_{\boldsymbol{i}}) \xrightarrow{u_{\boldsymbol{r}}} \mathcal{M}_{style} \xrightarrow{u_{\boldsymbol{r}}} \mathcal{M}_{acoustic}$
- **Vevo-TTS** for voice imitation (synthesis task): $\widetilde{\boldsymbol{Q}_c}(\mathcal{T}_{\boldsymbol{i}}) \xrightarrow{u_{\boldsymbol{r}}} \widetilde{\mathcal{M}}_{style} \xrightarrow{u_{\boldsymbol{r}}} \mathcal{M}_{acoustic}$

For Vevo-TTS, $\widetilde{\boldsymbol{Q}_c}(\mathcal{T}_{\boldsymbol{i}})$ means the tokenization for $\mathcal{T}_{\boldsymbol{i}}$, and $\widetilde{\mathcal{M}}_{style}$ means the pre-trained model for content-style modeling that takes text as input. We describe its detailed design in Appendix B.3.

## 4 Experiments

**Training Data**  We train the English-only models on 60K hours of ASR-transcribed English audiobooks, which is the same as the dataset used by the Voicebox English model [27]. The model $\mathcal{M}_{acoustic}$ and $\mathcal{M}_{style}$ are trained solely with speech data. The model $\widetilde{\mathcal{M}}_{style}$, which uses text as input, is trained with both speech and textual transcriptions data. We begin with the publicly available HuBERT-Large[1] model [2] to prepare the VQ-VAE tokenizer. We utilize its hidden features from

---

[1] https://pytorch.org/audio/0.10.0/pipelines.html#hubert-large

Table 2: Performance of $\mathcal{M}_{acoustic}$ trained by different HuBERT representations on zero-shot timbre imitation task. We highlight three key turning points during the self-supervised disentanglement process: the **initial stage** of information filtering (the 18th layer features, where vocabulary size can be considered infinite), the proposed **content-style tokenizer** (VQ-VAE tokens with a vocabulary size of 4096), and the proposed **content tokenizer** (VQ-VAE tokens with a vocabulary size of 32).

| Representations | #Vocab | WER ($\downarrow$) | S-SIM (to ref) ($\uparrow$) | S-SIM (to src) ($\downarrow$) | FPC (to src) ($\uparrow$) | Analysis |
|---|---|---|---|---|---|---|
| Ground Truth | - | 5.526 | 0.762 | 0.087 | 1.000 | - |
| 24th layer features | $\infty$ | 5.706 | 0.266 | 0.400 | 0.768 | **Pros**: Intelligibility, Style consistency |
| 18th layer features | $\infty$ | 5.324 | 0.250 | 0.505 $\uparrow$ | 0.824 | **Cons**: Timbre imitation |
| 12th layer features | $\infty$ | 5.348 | 0.200 | 0.626 $\uparrow$ | 0.805 | |
| PPG features | $\infty$ | 6.143 | 0.449 | 0.157 | 0.741 | **Pros**: Intelligibility, Timbre imitation |
| ASR tokens | 29 | 7.836 | 0.463 | 0.125 | 0.698 | **Cons**: Style consistency |
| K-means tokens | 1024 | 11.493 | 0.398 | 0.150 | 0.734 | Worse than VQ-VAE tokens (1024) |
| VQ-VAE tokens | 16384 | 6.807 | 0.398 | 0.306 | 0.826 | As the vocabulary size decreases, |
| | 4096 | 6.908 $\uparrow$ | 0.403 | 0.236 $\downarrow$ | 0.797 $\downarrow$ | **Pros**: |
| | 1024 | 6.967 $\uparrow$ | 0.418 | 0.249 | 0.764 $\downarrow$ | Timbre imitation $\uparrow$ |
| | 32 | 9.731 $\uparrow$ | 0.426 | 0.161 $\downarrow$ | 0.706 $\downarrow$ | **Cons**: |
| | 16 | 13.169 $\uparrow$ | 0.441 | 0.146 $\downarrow$ | 0.672 $\downarrow$ | Intelligibility $\downarrow$ |
| | 8 | 21.813 $\uparrow$ | 0.392 | 0.109 $\downarrow$ | 0.675 | Style consistency $\downarrow$ |

* PPG features and ASR tokens are obtained from HuBERT-ASR-Large, while the others are from HuBERT-Large. K-means and VQ-VAE tokens are quantized on the 18th layer features of HuBERT-Large. FPC are evaluated only on EMOTION.
* **#Vocab**: the vocabulary size $K$. **S-SIM**: Speaker SIM. ref/src: reference/source.

the 18th layer as the reconstruction objective for the tokenizer. Both the content and content-style tokenizers are trained on a 100-hour subset randomly sampled from the full 60K-hour dataset.

**Evaluation Data** We consider various evaluation settings to construct the evaluation set: (1) For clean data, such as recordings made in studio environments, we select audiobook speech data. Specifically, we reserve a subset of the total 60K hours of data as evaluation samples, which we denote as AB. (2) For noisy data, which may include in-the-wild recordings and diverse recording devices, we use the Common Voice English dataset (CV) [68]. It covers broader accents and is noisier compared to AB. (3) Additionally, to introduce more stylized and expressive data, we use an internal emotional and accented corpus to sample an emotional test set (EMOTION) and an accented test set (ACCENT). There are 700 evaluation samples in total: 200 from AB, 200 from CV, 150 from ACCENT, and 150 from EMOTION.

**Evaluation Metrics** For the objective metrics, we evaluate the intelligibility (WER), speaker similarity (S-SIM), accent similarity (A-SIM), emotion similarity (E-SIM), and F0 correlation (FPC) [44, 69]. Specially, we calculate WER based on Whisper-large-v3 [11, 13, 70]. For the three similarity metrics – S-SIM, A-SIM, and E-SIM – we calculate the cosine similarity between the embeddings (of speaker, accent, or emotion) of the generated sample and the reference. Specifically, we extract these embeddings using WavLM TDNN[2] [11, 13, 64] for speaker, CommonAccent[3] [21, 71] for accent, and emotion2vec[4] [72] for emotion, respectively. We also used CommonAccent and emotion2vec as the classifiers to measure the classification accuracy of accent and emotion (A-ACC and E-ACC). For subjective metrics, we use the Mean Opinion Score (MOS, rated from 1 to 5) to assess naturalness (N-MOS) and similarity in speaker, accent, emotion, and prosody (SS-MOS, AS-MOS, ES-MOS, and PS-MOS). SS MOS, AS-MOS, and ES-MOS evaluate the similarity between the generated sample and the *reference*, while PS-MOS assesses the similarity between the generated sample and the *source*. Additionally, we employ Comparative MOS (CMOS, rated from -3 to 3) to evaluate naturalness (N-CMOS), accentedness (A-CMOS), and emotiveness (E-CMOS). Detailed backgrounds of subjects and definitions of all subjective metrics are provided in Appendix E.

## 4.1 EFFECT OF THE VOCABULARY SIZE OF THE VQ-VAE TOKENIZER

We conduct experiments to figure out how to derive *content* and *content-style* tokens from speech. The key questions include: (1) What information from speech is retained in the *continuous hidden*

---

[2]https://github.com/microsoft/UniSpeech/tree/main/downstreams/speaker_verification

[3]https://huggingface.co/Jzuluaga/accent-id-commonaccent_ecapa

[4]https://github.com/ddlBoJack/emotion2vec

Table 3: Results on zero-shot timbre imitation and voice imitation (conversion) tasks. (ContRep/Model: Hours of training data for the used content representations and the model)

| *Zero-Shot Timbre Imitation* (AB, CV) | | | | | Source $i$, Reference $r$ $\Rightarrow$ Target: $\mathcal{W}(c_i, s_i, t_r)$ | | | |
|---|---|---|---|---|---|---|---|---|
| **Model** | **AR?** | **Training Data** (ContRep / Model) | **WER** ($\downarrow$) | **S-SIM** (to $r$) ($\uparrow$) | **FPC** (to $i$) ($\uparrow$) | **N-MOS** ($\uparrow$) | **SS-MOS** (to $r$) ($\uparrow$) | **PS-MOS** (to $i$) ($\uparrow$) |
| HierSpeech++ [53] | ✗ | 500K / 2.8K | 4.233 | 0.385 | 0.634 | 3.05 $_{\pm 0.23}$ | 3.24 $_{\pm 0.25}$ | 3.08 $_{\pm 0.26}$ |
| LM-VC [52] | ✓ | 1K / 60K | 8.623 | 0.310 | 0.524 | 2.90 $_{\pm 0.11}$ | 2.98 $_{\pm 0.18}$ | 2.16 $_{\pm 0.26}$ |
| UniAudio [28] | ✓ | 1K / 100K | 7.241 | 0.264 | 0.575 | 3.04 $_{\pm 0.15}$ | 2.47 $_{\pm 0.20}$ | 2.51 $_{\pm 0.25}$ |
| FACodec [17] | ✗ | 60K / 60K | 3.682 | 0.327 | 0.611 | 2.50 $_{\pm 0.20}$ | 3.10 $_{\pm 0.24}$ | 3.10 $_{\pm 0.23}$ |
| Vevo-Voice | ✓ | 60K / 60K | 7.694 | **0.458** | 0.485 | 3.09 $_{\pm 0.13}$ | **3.51** $_{\pm 0.24}$ | 2.60 $_{\pm 0.23}$ |
| Vevo-Timbre | ✗ | 60K / 60K | **2.968** | 0.420 | **0.686** | **3.35** $_{\pm 0.09}$ | 3.36 $_{\pm 0.16}$ | **3.45** $_{\pm 0.17}$ |

| *Zero-Shot Voice Imitation* (ACCENT, EMOTION) | | | | Source $i$, Reference $r$ $\Rightarrow$ Target: $\mathcal{W}(c_i, s_r, t_r)$ | | | |
|---|---|---|---|---|---|---|---|
| **Model** | **WER** ($\downarrow$) | **S-SIM** (to $r$) ($\uparrow$) | **A-SIM** (to $r$) ($\uparrow$) | **E-SIM** (to $r$) ($\uparrow$) | **N-MOS** ($\uparrow$) | **SS-MOS** (to $r$) ($\uparrow$) | **AS-MOS** (to $r$) ($\uparrow$) | **ES-MOS** (to $r$) ($\uparrow$) |
| Ground Truth | 10.917 | 0.762 | 0.763 | 0.965 | - | - | - | - |
| HierSpeech++ [53] | 12.921 | 0.466 | 0.526 | 0.658 | 3.04 $_{\pm 0.14}$ | 3.15 $_{\pm 0.23}$ | 3.13 $_{\pm 0.22}$ | 2.55 $_{\pm 0.19}$ |
| LM-VC [52] | 20.353 | 0.312 | 0.426 | 0.649 | 2.40 $_{\pm 0.10}$ | 2.56 $_{\pm 0.15}$ | 3.02 $_{\pm 0.19}$ | 2.46 $_{\pm 0.17}$ |
| UniAudio [28] | 15.751 | 0.311 | 0.486 | 0.611 | 2.95 $_{\pm 0.11}$ | 2.39 $_{\pm 0.17}$ | 2.42 $_{\pm 0.15}$ | 2.41 $_{\pm 0.26}$ |
| FACodec [17] | 12.731 | 0.434 | 0.514 | 0.688 | 2.36 $_{\pm 0.18}$ | 3.19 $_{\pm 0.22}$ | 3.01 $_{\pm 0.16}$ | 2.30 $_{\pm 0.22}$ |
| Vevo-Timbre | **12.351** | 0.486 | 0.567 | 0.816 | **3.43** $_{\pm 0.09}$ | 3.46 $_{\pm 0.15}$ | 3.55 $_{\pm 0.25}$ | 2.66 $_{\pm 0.26}$ |
| Vevo-Voice | 15.214 | **0.517** | **0.614** | **0.872** | 3.24 $_{\pm 0.11}$ | **3.70** $_{\pm 0.24}$ | **3.90** $_{\pm 0.19}$ | **3.20** $_{\pm 0.16}$ |

[1] PS-MOS, E-SIM, and ES-MOS are evaluated only on EMOTION. A-SIM and AS-MOS are evaluated only on ACCENT.
[2] The best and the second best result is shown in **bold** and by underlined.

*features* of HuBERT? (2) How do vector quantization methods, including the commonly used K-means [2, 28, 52, 73] and our adopted VQ-VAE [1, 62], affect the disentanglement ability of the resulting *discrete tokens* of HuBERT? (3) How does the vocabulary size of VQ-VAE codebook influence the produced tokens? To answer these questions, we investigate the performance of different HuBERT representations on the zero-shot timbre imitation task – i.e., using them to train $\mathcal{M}_{acoustic}$.

Specifically, we adopt the representations of the HuBERT-Large[1] model, which is a 24-layer transformer pre-trained on Libri-light dataset [74]. For comparison, we also examine the HuBERT-ASR-Large[5] model, which is fine-tuned from HuBERT-Large for ASR task on LibriSpeech [75]. Compared to HuBERT-Large, HuBERT-ASR-Large contains an additional prediction layer and a softmax layer, whose output is $x_{ppg} \in \mathbb{R}^{T \times 29}$, where $T$ is the frame length and 29 represents the vocabulary size of phonemes. We refer to $x_{ppg}$ as PPG features and also derive frame-level ASR tokens from each frame's PPG features: $x_{asr} = \arg\max x_{ppg} \in \mathbb{R}^T$. We randomly sample a 6K-hour subset from the full training data for training. The results are presented in Table 2.

Our findings indicate that: (1) HuBERT continuous hidden features possess rich information on timbre (high S-SIM to source), style (high FPC), and linguistic content (low WER). Notably, the S-SIM to source is even higher than that to reference, i.e., there is a timbre leakage. This phenomenon is more obvious for shallower 12th layer features. (2) After ASR fine-tuning, both PPG features and ASR tokens retain substantial linguistic content information (low WER) but exhibit a significant reduction in timbre information (lower S-SIM to source) and a decrease in style information (lower FPC). (3) Compared to VQ-VAE, K-means tokens show lower intelligibility, S-SIM to reference, and FPC when $K$ is the same (1024). Huang et al. provides a detailed comparison of between these two methods recently [62]. (4) For VQ-VAE tokens, larger vocabulary sizes (e.g., 16384) retain more timbre information (S-SIM to source at 0.306). As $K$ decreases to 4096, much of the timbre information is filtered out (S-SIM to source/reference at 0.236/0.403), yet style information is relatively retained (FPC at 0.797). When $K$ reduces further to 32, in addition to timbre, most style information is also filtered out – FPC drops to 0.706, similar to ASR tokens. As $K$ diminishes to 16 or even 8, even the high-level linguistic content begins to be filtered out (rapid increase in WER).

Based on these findings, we select VQ-VAE with $K_c = 32$ for the content tokenizer and $K_s = 4096$ for the content-style tokenizer. Note that designing the information bottleneck is a challenging trade-off, and such $K_c$ and $K_s$ may not be optimal. However, the results in the following sections show that such a choice has been pretty good under various voice imitation tasks. Additional effects of different $(K_c, K_s)$ combinations on $\mathcal{M}_{style}$ training are detailed in Appendix D.1.

---

[5] https://pytorch.org/audio/0.10.0/pipelines.html#hubert-asr-large

Table 4: Results on style imitation task. (**PC**: Parallel corpus. **SL**: Style labels)

| Model | Zero-shot | Supervision | | | WER (↓) | A- / E-ACC (↑) | A- / E-SIM (↑) | N-COMS (↑) | A- / E-CMOS (↑) |
|---|---|---|---|---|---|---|---|---|---|
| | | PC | SL | Text | | | | | |
| ASR-AC [29] | ✗ | ✗ | ✓ | ✓ | 4.775 | 0.633 | - | 0.00 | 0.00 |
| Vevo-Style (ASR) | ✓ | ✗ | ✗ | ✓ | **1.550** | **0.723** | **0.570** | **0.32** $_{\pm 0.11}$ | **0.49** $_{\pm 0.14}$ |
| Vevo-Style | ✓ | ✗ | ✗ | ✗ | 3.083 | 0.663 | 0.562 | 0.30 $_{\pm 0.13}$ | 0.35 $_{\pm 0.21}$ |
| VoiceShop [20] | ✗ | ✓ | ✓ | ✓ | 5.547 | 0.642 | - | 0.00 | 0.00 |
| Vevo-Style (ASR) | ✓ | ✗ | ✗ | ✓ | **3.553** | **0.735** | **0.585** | **0.26** $_{\pm 0.16}$ | **0.18** $_{\pm 0.20}$ |
| Vevo-Style | ✓ | ✗ | ✗ | ✗ | 5.464 | 0.673 | 0.554 | 0.12 $_{\pm 0.10}$ | 0.13 $_{\pm 0.08}$ |
| Conv-Speak [21] | ✗ | ✓ | ✓ | ✗ | 9.950 | 0.571 | - | 0.00 | 0.00 |
| Vevo-Style (ASR) | ✓ | ✗ | ✗ | ✓ | **2.778** | 0.864 | 0.574 | 0.10 $_{\pm 0.05}$ | 0.40 $_{\pm 0.12}$ |
| Vevo-Style | ✓ | ✗ | ✗ | ✗ | 3.889 | **0.903** | **0.580** | **0.15** $_{\pm 0.12}$ | **0.60** $_{\pm 0.16}$ |
| Emovox [30] | ✗ | ✗ | ✓ | ✓ | 15.444 | 0.750 | - | 0.00 | 0.00 |
| Vevo-Style (ASR) | ✓ | ✗ | ✗ | ✓ | **9.842** | 0.692 | 0.800 | 1.74 $_{\pm 0.20}$ | 0.45 $_{\pm 0.11}$ |
| Vevo-Style | ✓ | ✗ | ✗ | ✗ | 10.221 | **0.754** | **0.825** | **1.78** $_{\pm 0.20}$ | **0.49** $_{\pm 0.13}$ |

\* We present four comparative groups. Evaluation samples for each group are sourced from the baseline's demo website. For the first three groups, we evaluate A-ACC/SIM/CMOS, and for the last group, we evaluate E-ACC/SIM/CMOS.

## 4.2 ZERO-SHOT TIMBRE IMITATION AND VOICE IMITATION (CONVERSION TASK)

Further, we apply Vevo to various zero-shot imitation tasks. This section evaluates **Vevo-Timbre** and **Vevo-Voice** on zero-shot timbre and voice imitation tasks. We select several state-of-the-art (SOTA) baselines in zero-shot voice conversion, including HierSpeech++ [53], LM-VC [52], UniAudio [28], and FACodec [17]. Details about these baselines are available in Appendix C. We train $\mathcal{M}_{style}$ and $\mathcal{M}_{acoustic}$ on the full 60K hours dataset. The results are presented in Table 3.

The findings reveal that: (1) **Zero-shot timbre imitation**: Compared to the four baselines, Vevo-Timbre exhibits superior performance across common voice conversion metrics such as WER, S-SIM, N-MOS, and SS-MOS. Additionally, Vevo-Timbre demonstrates a clear advantage in FPC and PS-MOS, which measure style consistency. (2) **Zero-shot voice imitation**: Against the four baselines, Vevo-Voice not only excels in mimicking speaker identity (S-SIM, SS-MOS) but also significantly outperforms in imitating specific style attributes like accent (A-SIM, AS-MOS) and emotion (E-SIM, ES-MOS). (3) **Comparing Vevo-Timbre and Vevo-Voice**: Vevo-Timbre's strength lies in preserving the style of the source (FPC, PS-MOS), whereas Vevo-Voice additionally excels in style imitation, resulting in higher speaker similarity (S-SIM, SS-MOS). However, due to the autoregressive design in $\mathcal{M}_{style}$, Vevo-Voice scores lower in intelligibility (WER) compared to Vevo-Timbre.

## 4.3 ZERO-SHOT STYLE IMITATION

We present the performance of **Vevo-Style** in the zero-shot style imitation task, focusing on widely studied styles such as accent and emotion. For accent imitation, we select baselines including ASR-AC [29], VoiceShop [20], and Conv-Speak [21]. We use their demo website samples as our evaluation set, including conversions among multiple accented English such as British, American, Hindi, and Mandarin. For emotion imitation, we choose Emovox [30] and its demo website samples, which include conversions from Neutral to Happy, Angry, and Sad emotions. Moreover, we also introduce the Vevo-Style (ASR) for a *zero-shot* style imitation baseline. Its only difference compared to the Vevo-Style model is the use of $x_{asr}$ (see Section 4.1) rather than $Q_c$ as the content tokenizer.

Our experimental results are shown in Table 4. Our observations indicate: (1) Compared to the baselines, Vevo-Style is only self-supervised trained on audiobook speech data. However, without any fine-tuning on accented or emotional corpus, it delivers superior outcomes in terms of intelligibility (WER), quality (N-CMOS), and the imitation of accents and emotions (A-ACC/SIM/CMOS and E-ACC/SIM/CMOS). (2) Using text as the additional supervision, Vevo-Style (ASR) further surpasses Vevo-Style in intelligibility and specific aspects of accent imitation. We speculate that compared to $x_{asr}$, the $Q_c$ used by Vevo-Style may still retain a small portion of accent-related information, thereby limiting the $\mathcal{M}_{style}$ to perfectly imitate the accent information from the style reference.

## 4.4 ZERO-SHOT VOICE IMITATION (SYNTHESIS TASK)

We present the performance of **Vevo-TTS** in the zero-shot voice imitation (synthesis) task. We select the classic baselines of the zero-shot TTS filed, including the Non-AR Voicebox model [27], and

Table 5: Results on zero-shot voice imitation (synthesis) task.

| Model | AR? | Training Data | WER ($\downarrow$) | S-SIM ($\uparrow$) | A-SIM ($\uparrow$) | E-SIM ($\uparrow$) | N-CMOS ($\uparrow$) | SS-MOS ($\uparrow$) | AS-MOS ($\uparrow$) | ES-MOS ($\uparrow$) |
|---|---|---|---|---|---|---|---|---|---|---|
| Ground Truth | - | - | 11.348 | 0.710 | 0.633 | 0.936 | 0.00 | - | - | - |
| CosyVoice [24] | ✓ | 171K | 8.400 | 0.614 | 0.640 | 0.839 | -0.18 $_{\pm 0.19}$ | 4.11 $_{\pm 0.19}$ | 3.99 $_{\pm 0.23}$ | 3.66 $_{\pm 0.19}$ |
| MaskGCT [13] | ✗ | 100K | 9.442 | 0.659 | 0.645 | 0.822 | -0.04 $_{\pm 0.19}$ | 4.16 $_{\pm 0.16}$ | 4.38 $_{\pm 0.14}$ | 3.76 $_{\pm 0.25}$ |
| VALL-E [26] | ✓ | 45K | 13.226 | 0.400 | 0.485 | 0.735 | -1.24 $_{\pm 0.42}$ | 2.82 $_{\pm 0.40}$ | 2.77 $_{\pm 0.45}$ | 2.63 $_{\pm 0.36}$ |
| Voicebox [27] | ✗ | 60K | **9.414** | 0.463 | 0.575 | 0.811 | -0.35 $_{\pm 0.21}$ | 3.87 $_{\pm 0.21}$ | 3.49 $_{\pm 0.29}$ | 3.61 $_{\pm 0.19}$ |
| VoiceCraft [77] | ✓ | 9K | 13.057 | 0.392 | 0.517 | 0.788 | -0.50 $_{\pm 0.23}$ | 3.47 $_{\pm 0.32}$ | 3.29 $_{\pm 0.28}$ | 3.52 $_{\pm 0.25}$ |
| Vevo-TTS | ✓ | 60K | 12.066 | **0.505** | **0.579** | **0.840** | **-0.14** $_{\pm 0.18}$ | **4.05** $_{\pm 0.21}$ | **4.12** $_{\pm 0.21}$ | **4.03** $_{\pm 0.19}$ |

[1] A-SIM and AS-MOS are evaluated on ACCENT samples. E-SIM and ES-MOS are evaluated on EMOTION samples.
[2] The best and the second best results of only the last four are shown in **bold** and by underlined.

Table 6: Effect of duration reduction and different inference modes of $\mathcal{M}_{style}$. (#Inference Input: input sequence length (%) during inference. w/o: without. w/: with)

| Model | #Inference Input | WER ($\downarrow$) | S-SIM ($\uparrow$) | A-SIM ($\uparrow$) | E-SIM ($\uparrow$) | DDUR ($\downarrow$) |
|---|---|---|---|---|---|---|
| Vevo-Voice | 100% | 15.214 | 0.517 | 0.614 | 0.883 | 0.933 |
| w/o *Duration Reduction* | 127% | 15.958 | 0.501 | 0.583 | 0.842 | 1.698 |
| w/ *Global-guided* continuation | 42% | 16.809 | 0.510 | 0.597 | 0.864 | 0.947 |

[*] Vevo-Voice uses reference-style-enhanced continuation. A-SIM is evaluated only on ACCENT samples. E-SIM is evaluated only on EMOTION samples. The remaining metrics are evaluated on both ACCENT and EMOTION samples.

the AR models such as VALL-E [26, 76] and VoiceCraft [77], all of which are trained only on audiobook speech data. For comparison, we also include two stronger SOTA models: CosyVoice [24] and MaskGCT [13, 78], which are trained on large-scale private corpus derived from in-the-wild video data, featuring highly diverse distributions [24, 79]. Detailed baseline information and evaluation results are available in Appendix D.2. Here, we only highlight performances on ACCENT and EMOTION evaluation samples (Table 5). Our observations are as follows: (1) Compared between Voicebox and Vevo-TTS whose training data are identical, Vevo-TTS, while showing slightly inferior performance in WER (which is a common weakness for AR models), excels across all other metrics. (2) Notably, Vevo-TTS demonstrates outstanding performance in style imitation (A/E-SIM, AS/ES-MOS). Despite being trained only on audiobook data, it surpasses CosyVoice and MaskGCT in some emotion imitation tasks (ES-MOS is 4.03). This verifies the effectiveness of our proposed content-style tokens, which could be representations that can effectively capture style information and are easily learned by downstream models.

## 4.5 EFFECT OF DURATION REDUCTION AND DIFFERENT INFERENCE MODES

Finally, we conduct ablation studies on several key components within Vevo, including the impact of different $(K_c, K_s)$ values on voice imitation tasks (see Appendix D.1), the effects of the duration reduction strategy, and the two inference modes of $\mathcal{M}_{style}$, as presented in Table 6. We adopt DDUR to measure the average differences in duration (seconds) between the converted and ground truth utterances [30, 51]. We observe that: (1) The duration reduction not only reduces the inference input length but also consistently demonstrates clear advantages, especially in duration conversion (DDUR). (2) The reference-global-guided continuation significantly shortens the sequence length (to 42% of Vevo-Voice), with only a slight decline in performance metrics. This showcases its substantial potential in saving inference memory and enhancing inference speed.

## 5 CONCLUSION

We introduce Vevo, a versatile zero-shot voice imitation framework featuring controllable timbre and style. Vevo contains of two primary stages: content-style modeling via an autoregressive transformer, and acoustic modeling via a flow matching transformer. Both stages are trainable through self-supervised and in-context learning, friendly to scale up. Vevo operates based on our newly proposed content and content-style tokens, generated by VQ-VAE tokenizers of HuBERT with carefully adjusted vocabulary sizes. Pre-trained only on 60K hours of audiobook speech data without fine-tuning on style-specific corpus, Vevo outperforms state-of-the-art models of accent and emotion conversion fields, particularly achieving these conversions in a zero-shot manner. Furthermore, Vevo's robust performance in zero-shot voice conversion and text-to-speech tasks underscores its versatility and also highlights the broad potential of our proposed disentangled speech tokens.

ACKNOWLEDGEMENT

This work is partially supported by the NSFC under Grant 62376237, Shenzhen Science and Technology Program ZDSYS20230626091302006. We thank Yuancheng Wang, Jilong Wu, Fuchun Peng, and the anonymous reviewers for their insightful comments and suggestions. We appreciate the efforts of Meiyu Zheng and all the subjects during the subjective evaluation.

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

# A TERMINOLOGY CLARIFICATION

In this study, we decouple speech into linguistic content (*what to speak*), timbre (*who speaks*), and style (*how to speak*). Below, we will clarify our definitions and scope for timbre and style.

**Timbre** Timbre is a *physical concept* that refers to the acoustic qualities of sound, such as the spectral envelope, which allows us to differentiate between speakers even when pitch and loudness are identical. It is primarily determined by the speaker's vocal anatomy and articulatory behaviors. Often discussed alongside timbre is speaker identity. Speaker identity is a *perceptual concept* – it encompasses not only timbre but also habitual speech patterns, idiosyncrasies, and other personal styles that make a speaker recognizable. While timbre lays the acoustic foundation of identity, speaker identity reflects the broader auditory impression formed by a listener.

**Style** Style refers to the expressive aspects of speech, including accent, emotion, and speaking habits, which dictate *how something is said*. It includes specific features such as *accent* and *emotion*, but also covers a wider array of expressive behaviors. A critical component of style is *prosody*, which includes features such as F0 (pitch), energy, and duration. These prosodic features govern the rhythm, stress, and intonation of speech, contributing significantly to how emotion and emphasis are conveyed. Although style encompasses prosody, it also extends beyond it, influencing not only the melodic flow of speech but also cultural and emotional expressions.

# B DETAILS OF VEVO

## B.1 VQ-VAE ARCHITECTURE

We adopt the implementation of RepCodec[6] [62] as our VQ-VAE tokenizer, whose $\lambda$ and $\beta$ are 45 and 1. Its architecture of encoder and decoder is shown in Figure 4. The vocabulary sizes of our *content* and *content-style* tokenizer are 32 and 4096. Their parameter counts are 59M and 63M, respectively.

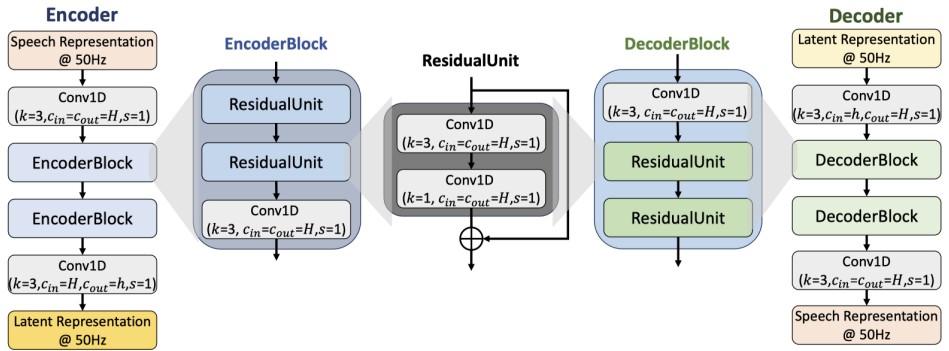

Figure 4: Encoder and decoder architecture of our VQ-VAE tokenizer. $k$, $s$. $c_{in}$, and $c_{out}$ denote the kernel size, stride, input channels, and output channels. $h$ denotes the vocabulary size of tokenizer. $H$ denotes the hidden dimension of input representations (which is 1024 for HuBERT-Large). This figure is borrowed from the paper of RepCodec [62].

## B.2 CONTENT-STYLE MODELING

For the content-style modeling stage, we use reference-style-enhanced continuation by default. The architecture of our AR transformer is similar to LLaMA[7] [35]. It has 12 layers, 16 attention heads, 2048/3072 embedding/feed-forward network (FFN) dimension. The global style encoder consists of WavLM-based representation layers and TDNN-based feature extraction layers [64, 65]. Specif-

---

[6]https://github.com/mct10/RepCodec
[7]https://github.com/meta-llama/llama3

ically, we adopt the same architecture with a WavLM-based speaker verification model[8]. The total parameter count of $\mathcal{M}_{style}$ is 463M.

During training, we use AdamW [80] optimizer with a peak learning rate of 1e-4, linearly warmed up for 2K steps and decays over the rest of training. It is trained for 500K updates. During inference, we can use the default reference-style-enhanced continuation (Figure 2b) or the reference-global-guided continuation (Figure 5). We generate evaluation samples with specific sampling parameters: top-k is 25, top-p is 0.9, and temperature is 0.8.

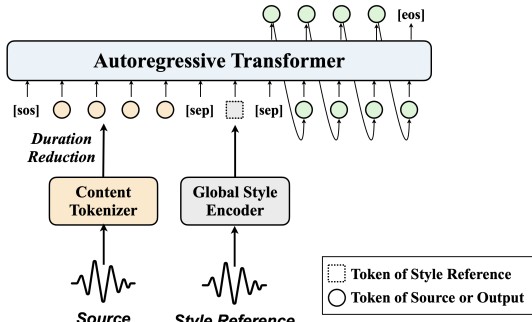

Figure 5: *Reference-global-guided* continuation of $\mathcal{M}_{style}$ for inference.

### B.3 CONTENT-STYLE MODELING (TEXT AS INPUT)

Compared to $\mathcal{M}_{style}$, the only difference of $\widetilde{\mathcal{M}}_{style}$ is that its input becomes text tokens, rather than the duration reduced content tokens. Specifically, we adopt the Grapheme-to-Phoneme (G2P) method and use the same phonemization tokenizer as Voicebox [27]. All the hyper parameters of training and inference are same as $\mathcal{M}_{style}$.

### B.4 ACOUSTIC MODELING

For the acoustic modeling stage, we follow the flow matching model implementation of Voicebox [27]. Specifically, we randomly mask 70%-100% of the frames to create $y_1^{mis}$. We employ the midpoint ODE solver with a step size of 0.0625 (NFE=32). The $\sigma$ of the optimal transport path of flow matching is 1e-5. The transformer has 24 layers, 16 attention heads, 1024/4096 embedding/feed-forward network (FFN) dimension. Its parameter count is 334M.

Our target Mel spectrogram is at 24 kHz with 100 Mel bands. It is normalized with the global mean (-5.8843) and standard deviation (2.2615) to stabilize training [27]. During training, Mel spectrogram length is capped at 1,600 frames and chunked randomly if length exceeds. We use Adam [81] optimizer with a peak learning rate of 1e-4, linearly warmed up for 5K steps and decays over the rest of training. It is trained for 500K updates. During inference, we employ the midpoint ODE solver with a step size of 0.0625 (NFE=32).

We apply the classifier free guidance (CFG) [82] to improve the generation quality like other works [12, 24, 27]. Specifically, we randomly drop the conditions, i.e., $y_1^{ctx}$ and $Q_s(u)$, with a probability of $p_{uncond}$. During inference, the modified vector filed $f_t'$ becomes $f_t'(y_t, t, y_1^{ctx}, Q_s(u)) = (1 + \alpha) f_t(y_t, t, y_1^{ctx}, Q_s(u)) - \alpha f_t(y_t, t)$, where $\alpha$ is the strength of the guidance. In practice, we follow Voicebox and set $p_{uncond}$ as 0.2 and $\alpha$ as 0.7.

### B.5 VOCODER

We use BigVGAN [67] as vocoder to synthesis waveform from Mel spectrogram. We fine-tune from the official released checkpoint bigvgan_24khz_100band[9] using our 60K hours training data. Its parameter count is 112M.

---

[8]https://huggingface.co/microsoft/wavlm-base-plus-sv
[9]https://github.com/NVIDIA/BigVGAN

## C  DETAILS OF BASELINES

### C.1  ZERO-SHOT TIMBRE IMITATION AND VOICE IMITATION (CONVERSION TASK)

- **HierSpeech++** [53]: It utilizes MMS [83] (pretrained on 500K hours of data from over 1000 languages) to extract content features. It is designed based on the VITS architecture [84], and is trained on 2.8k hours sourced from Libri-light [74] and LibriTTS [85]. We use the officially released checkpoint[10] to generate samples.

- **LM-VC** [52]: It is an autoregressive hierarchical transformer that predicts SoundStream [63] codecs from soft units similar to HuBERT k-means tokens [54], trained on the Libri-light dataset [74]. We obtain the generated samples from the authors.

- **UniAudio** [28]: It is an autoregressive transformer capable of performing multiple audio generation tasks, using 500-cluster K-means tokens from HuBERT-base (that is pre-trained on LibriSpeech [75]) to predict their proposed acoustic codecs, with training data comprising approximately 80K hours of speech and 20K hours of other audio data. We use the officially released checkpoint[11] to generate samples.

- **FACodec** [17]: It adopts an auto-encoder and residual vector quantization based architecture. It decouples the raw waveform into factorized attributes through ASR, F0 prediction, and speaker classification tasks, trained on the Libri-light dataset [74]. We use the released checkpoint in Amphion [12] [76, 78] (which is implemented by the authors) to generate samples.

### C.2  ZERO-SHOT STYLE IMITATION

- **ASR-AC** [29]: It uses an ASR model based on wav2vec 2.0[13] [86] (that is pre-trained on 60K hours of Libri-light [74] and fine-tuned on 1K hours of LibriSpeech [75]) to extract the one-hot text predictions from speech, i.e., $x_{asr}$ in our paper (Section 4.1). It adopts a transformer encoder and a HiFi-GAN decoder to reconstruct waveforms conditioned on $x_{asr}$, the accent labels, and F0, which is trained on about 700 hours of accented corpus. We use 30 samples from its demo website[14] to evaluate, including English accents' conversions from British to American, British to Hindi, and Hindi to American.

- **VoiceShop** [20]: To achieve accent conversion, the authors first uses an conformer-based ASR model (that is trained by 40K hours of their private corpus) to extract the hidden features (BNF). Then, they create about 300 hours of parallel conversion corpus based on a commercial accented TTS system. Finally, they adopt an encoder-decoder transformer to learn the BNF's mapping between parallel corpus. We use 17 samples from its demo website[15] to evaluate, including English accents' conversions among American, British, Hindi, and Mandarin.

- **Conv-Speak** [21]: The authors formulate accent conversion from source's content tokens to target's content tokens. They propose to self-supervised pre-train on content tokens like BART [87], in order to relieve the requirements of parallel data. They adopt the 500-cluster K-means of HuBERT-Base[16] (that is pre-trained on 1K hours of LibriSpeech [75]) as content tokens. The conversion model is trained on about 600 hours of data, including about 1 hour of parallel data. We use 24 samples from its demo website[17] to evaluate, including English accents' conversions from Hindi and Mandarin to American.

- **Emovox** [30]: To achieve emotion conversion, the authors design a recognition encoder to push its output (i.e., emotion-agnostic features) closely with phoneme transcriptions. The conversion model is based on a sequence-to-sequence decoder, that can reconstruct the Mel spectrogram conditioned on the emotion-agnostic features and emotion labels. The model is trained on about

---

[10]https://github.com/sh-lee-prml/HierSpeechpp

[11]https://github.com/yangdongchao/UniAudio

[12]https://huggingface.co/amphion/naturalspeech3_facodec

[13]https://pytorch.org/audio/0.10.0/pipelines.html#torchaudio.pipelines.WAV2VEC2_ASR_LARGE_LV60K_960H

[14]https://accent-conversion.github.io/

[15]https://voiceshopai.github.io/

[16]https://pytorch.org/audio/0.10.0/pipelines.html#torchaudio.pipelines.HUBERT_BASE

[17]https://convert-and-speak.github.io/demo/

80 hours data sourced from VCTK [88] and ESD [7]. We use 24 samples from its demo website[18] to evaluate, including emotion conversions from Neutral to Angry, Happy, and Sad.

Notably, Vevo-Style (ASR) and Vevo-Style employ a zero-shot approach to achieve style imitation – which is rarely seen in existing research. Therefore, for these two, we use the aforementioned evaluation samples as the source and additionally prepare style references. Specifically, for accent imitation, we prepare references in three English accents: American, Hindi, and Mandarin, with four samples each (two males, and two females). For emotion imitation, we prepare references for three emotions: Angry, Happy, and Sad, with four samples each (two males, and two females). All these references are randomly sampled from ACCENT and EMOTION.

## C.3 Zero-Shot Voice Imitation (Synthesis Task)

- **VALL-E** [26]: It is a classic AR model for zero-shot TTS. It utilizes the transformer to predict EnCodec [89] codecs. We use the released checkpoint in Amphion [19] [76, 78] to generate samples, which is pre-trained on 45K hours of MLS English set [90].

- **Voicebox** [27]: It applies the flow matching transformer to both duration model and acoustic model. We reproduce it with the help of the authors.

- **VoiceCraft** [77]: It uses an AR transformer to predict EnCodec [89] codecs. Compared to VALL-E, it proposes token rearrangement and delayed stacking strategies to enhance the model learning. We use the officially released checkpoint[20] to generate samples, which is pre-trained on 10K hours of Gigaspeech [91].

- **CosyVoice** [24]: It proposes a semantic tokenizer that is supervised by ASR task. It contains an AR transformer to predict the semantic tokens from text, and a flow-matching transformer to predict Mel spectrograms. We use the officially released checkpoint[21] to generate samples, which is pre-trained on 171K hours of in-the-wild, multilingual, and private data.

- **MaskGCT** [13]: It consists of two-stage discrete diffusion models. It is based on the hidden features of w2v-bert 2.0[22] that pre-trained on 4.5M hours to obtain the semantic tokens. Its TTS model is trained on 100K hours of in-the-wild and multilingual data [79]. We use the released checkpoint in Amphion [23] [76, 78] to generate samples.

## D Additional Experimental results

### D.1 Effect of the Vocabulary Size of the VQ-VAE Tokenizer

In Section 4.1, we have already demonstrated the impact of different vocabulary sizes in the VQ-VAE codebook on $\mathcal{M}_{acoustic}$ (i.e., disentanglement capability). In this section, we aim to present two complementary experimental results. First, we explore the effects of a wider range of vocabulary sizes (from the smallest at 8 to the largest at 16,384) on the produced tokens. Second, we investigate the impact of various combinations of $K_c$ and $K_s$ on Vevo-Voice.

#### D.1.1 Effect on Phonetic Discriminability

We explore the phonetic discriminability of different representations, inspired by AudioLM [92]. Specifically, we measure phonetic discriminability using the ABX error rate, a distance-based metric that evaluates a set of phoneme trigrams differing only in the central phoneme (e.g., "bit" vs. "bet"). The ABX error rate assesses how often a random instance X of a trigram ("bit") is closer to an instance B of another trigram ("bet") rather than to another instance A of the same trigram ("bit").

---

[18]https://kunzhou9646.github.io/Emovox_demo/
[19]https://github.com/open-mmlab/Amphion/tree/main/egs/tts/VALLE_V2
[20]https://github.com/jasonppy/VoiceCraft
[21]https://github.com/FunAudioLLM/CosyVoice
[22]https://huggingface.co/facebook/w2v-bert-2.0
[23]https://huggingface.co/amphion/MaskGCT

We evaluate scenarios where all three sounds A, B, and X are uttered by the same speaker (within-speaker) and where the same speaker utters A and B but X comes from a different speaker (across-speaker). We calculate ABX using scripts provided with the Libri-light dataset[24] [74], employing default settings and reporting scores obtained on the LibriSpeech dev-clean dataset [75]. The results are displayed in Table 7. Note that for K-means and VQ-VAE tokens, we calculate the ABX error based on the centroid's vector corresponding to each token.

Table 7: ABX error rate ($\downarrow$) of different representations. (within/across speakers)

| Repr \ #Vocab | 8 | 16 | 32 | 64 | 128 | 256 | 512 | 1024 | 2048 | 4096 | 8192 | 16384 |
|---|---|---|---|---|---|---|---|---|---|---|---|---|
| PPG features | 6.1 / 7.0 | | | | | | | | | | | |
| 18th layer features | 7.6 / 9.5 | | | | | | | | | | | |
| K-means tokens | - | - | 17.2 / 19.8 | 14.5 / 17.7 | 12.0 / 14.0 | 9.9 / 11.5 | 8.8 / 10.3 | 7.8 / 9.0 | - | - | - | - |
| VQ-VAE tokens | 16.4 / 18.2 | 13.1 / 14.6 | 12.7 / 14.2 | 13.0 / 14.9 | 12.7 / 14.8 | 12.7 / 14.6 | 11.1 / 13.0 | 10.0 / 11.8 | 10.1 / 12.0 | 10.4 / 12.4 | 9.9 / 11.9 | 10.0 / 11.9 |

[*] PPG features are obatined from HuBERT-ASR-Large, while the others are from HuBERT-Large. K-means and VQ-VAE tokens are quantized on the 18th layer features of HuBERT-Large.

From the table, we observe that: (1) PPG features demonstrate the best phonetic discriminability, highlighting the advantages of fine-tuning with ASR tasks; (2) For K-means tokens, increasing the vocabulary size from 32 to 1024 continuously improves their phonetic discriminability, indicating an ongoing enhancement in their capacity to represent linguistic content; (3) For VQ-VAE tokens, we see a gradual improvement in phonetic discriminability from 8 to 1024, but beyond 1024, this metric begins to converge. However, we know that as VQ-VAE tokens' vocabulary size increases from 1024 to 4096 to 16384, their style information still increases (as indicated by rising FPC scores in Table 2). From these observations, we can conclude that beyond 1024, the representation ability of VQ-VAE tokens for linguistic content stabilizes, and any increase in vocabulary size is likely allocated to storing style information such as F0; (4) Comparing K-means and VQ-VAE tokens, it's evident that VQ-VAE tokens are less sensitive to changes in vocabulary size in terms of representing linguistic content (e.g., VQ-VAE (32) and K-means (128) exhibit nearly identical ABX error rates), suggesting that a smaller vocabulary can suffice for a content tokenizer. Recent research has also delved into this aspect, attributing the differences to the distinct optimization algorithms used by the two methods [62].

### D.1.2 EFFECT ON VEVO-VOICE

We explore the effects of different $(K_c, K_s)$ combinations on Vevo-Voice, with the results presented in Table 8. Our observations include: (1) A significant drop in intelligibility occurs when $K_c$ changes from 32 to 16, indicating that a smaller vocabulary size for the content tokenizer leads to loss of linguistic content information; (2) When $K_s$ decreases from 4096 to 1024, all metrics decline. We hypothesize that while a reduction in $K_s$ might lessen the learning difficulty for $\mathcal{M}_{style}$, a smaller $K_s$ also results in a decrease in the quality of the final generated audio for $\mathcal{M}_{acoustic}$.

Table 8: Effect of different $(K_c, K_s)$ for Vevo-Voice.

| Content Tokenizer ($K_c$) | Content-style Tokenizer ($K_s$) | WER ($\downarrow$) | S-SIM ($\uparrow$) | A-SIM ($\uparrow$) | E-SIM ($\uparrow$) |
|---|---|---|---|---|---|
| 32 | 4096 | **15.214** | **0.517** | **0.614** | **0.872** |
| 32 | 1024 | 18.523 | 0.502 | 0.609 | 0.860 |
| 16 | 4096 | 23.351 | 0.509 | 0.613 | 0.865 |

[*] Vevo-Voice uses $K_c$ as 32 and $K_s$ as 4096. E-SIM is evaluated only on EMOTION. A-SIM is evaluated only on ACCENT.

---

[24]https://github.com/facebookresearch/libri-light/tree/main/eval

## D.2 Zero-Shot Voice Imitation (Synthesis Task)

In Section 4.4, we present the performance of Vevo-TTS in zero-shot imitation (synthesis) tasks. We have detailed its comparative performance against baselines on all four evaluation sets (AB, CV, ACCENT, and EMOTION) in Table 9. We can observe that: (1) In comparison with AR baselines, Vevo-TTS exhibits a clear advantage over VALL-E and VoiceCraft across various metrics on all datasets. Compared to the state-of-the-art CosyVoice, although Vevo-TTS is trained solely on 60K hours of Audiobook data, it performs better in some metrics such as Naturalness CMOS (AB, AC-CENT, EMOTION), Speaker S-MOS (EMOTION), and notably in style imitation-related metrics like Accent S-MOS and Emotion S-MOS. This demonstrates the high effectiveness of the AR TTS model implemented using the content-style tokens proposed in this paper. (2) When compared with Non-AR baselines, Vevo-TTS falls short on WER across all datasets compared to Voicebox and MaskGCT. This underscores the stability still needed in AR models, indicating significant room for improvement.

# E Subjective Evaluation

## E.1 Background of Subjects

We hired dozens of subjects on a paid basis to complete the subjective evaluations. These individuals have extensive experience in providing subjective assessments of audio generated by AI models. They have lived in English-speaking countries for extended periods and are highly familiar with various common English accents, including American, British, Hindi, and Mandarin. Each audio sample in our evaluation was rated at least ten times.

## E.2 Metrics and Questionnaires

We have developed an automated subjective evaluation interface. For each item to be evaluated, users will see three components: the System Interface (i.e., the audio to be evaluated), the Questionnaire, and the Scoring Criteria.

### E.2.1 Naturalness MOS

**System Interface**  One audio to be evaluated (with target text)

**Questionnaire**  How human-like is the speech in the clip? Does it sound like a real human who is engaged in the topic, or does it sound like an AI that doesn't understand what is being said?

**Scoring criteria**  5 (A perfect imitation of human speech), 4 (Exceeds my expectations for AI voices), 3 (Meets my expectations for AI voices), 2 (A subpar representation of human speech), 1 (Very poor artificial speech)

### E.2.2 Speaker Similarity MOS

**System Interface**  One reference audio, One audio to be evaluated

**Questionnaire**  Ignore the content and audio quality, just pay attention to the voice of the person. How similar is the voice to be evaluated compared to the reference voice?

**Scoring criteria**  5 (Excellent, sounds like exactly the same person), 4 (Good, sounds like a similar person), 3 (Fair, sounds like a slightly similar person), 2 (Poor, sounds like a different person mostly), 1 (Bad, sounds like a completely different person)

### E.2.3 Accent Similarity MOS

**System Interface**  One reference audio, One audio to be evaluated

**Questionnaire**  Ignore the vocal characteristics (who is speaking), just pay attention to the accent of the speaker. Is the accent similar to the reference voice?

Table 9: Results on zero-shot imitation (synthesis) task. (S-MOS: Similarity MOS)

| Model | AR? | Training Data | WER (↓) | Speaker SIM (↑) | Naturalness CMOS (↑) | Speaker S-MOS (↑) |
|---|---|---|---|---|---|---|
| *AB* | | | | | | |
| Ground Truth | - | - | 2.845 | 0.763 | 0.00 | - |
| CosyVoice [24] | ✓ | 171k hours, In-the-wild | 3.647 | 0.727 | -0.44 $_{\pm0.16}$ | 4.17 $_{\pm0.15}$ |
| MaskGCT [13] | ✗ | 100K hours, In-the-wild | 3.841 | 0.781 | -0.21 $_{\pm0.08}$ | 4.30 $_{\pm0.22}$ |
| VALL-E [26] | ✓ | 45K hours, MLS English [90] | 8.204 | 0.551 | -0.95 $_{\pm0.39}$ | 3.27 $_{\pm0.25}$ |
| Voicebox [27] | ✗ | 60K hours, Audiobook | **3.175** | **0.631** | -0.55 $_{\pm0.15}$ | 3.49 $_{\pm0.14}$ |
| VoiceCraft [77] | ✓ | 10K hours, Gigaspeech [91] | 4.737 | 0.570 | -0.41 $_{\pm0.18}$ | 3.41 $_{\pm0.13}$ |
| Vevo-TTS | ✓ | 60K hours, Audiobook | 3.672 | 0.593 | **-0.31** $_{\pm0.14}$ | **3.58** $_{\pm0.15}$ |
| *CV* | | | | | | |
| Ground Truth | - | - | 1.426 | 0.723 | 0.00 | - |
| CosyVoice [24] | ✓ | 171k hours, In-the-wild | 3.500 | 0.627 | 0.11 $_{\pm0.19}$ | 3.88 $_{\pm0.12}$ |
| MaskGCT [13] | ✗ | 100K hours, In-the-wild | 2.573 | 0.688 | 0.08 $_{\pm0.25}$ | 4.33 $_{\pm0.14}$ |
| VALL-E [26] | ✓ | 45K hours, MLS English [90] | 6.129 | 0.433 | -1.02 $_{\pm0.36}$ | 2.68 $_{\pm0.52}$ |
| Voicebox [27] | ✗ | 60K hours, Audiobook | **2.129** | 0.500 | -0.12 $_{\pm0.19}$ | 2.91 $_{\pm0.08}$ |
| VoiceCraft [77] | ✓ | 10K hours, Gigaspeech [91] | 6.353 | 0.446 | **-0.10** $_{\pm0.25}$ | 3.02 $_{\pm0.21}$ |
| Vevo-TTS | ✓ | 60K hours, Audiobook | 2.687 | **0.513** | -0.11 $_{\pm0.19}$ | **3.83** $_{\pm0.18}$ |

| Model | AR? | WER (↓) | Speaker SIM (↑) | Accent SIM (↑) | Naturalness CMOS (↑) | Speaker S-MOS (↑) | Accent S-MOS (↑) |
|---|---|---|---|---|---|---|---|
| *ACCENT* | | | | | | | |
| Ground Truth | - | 10.903 | 0.747 | 0.633 | 0.00 | - | - |
| CosyVoice [24] | ✓ | 6.660 | 0.653 | 0.640 | 0.10 $_{\pm0.19}$ | 4.23 $_{\pm0.18}$ | 3.99 $_{\pm0.23}$ |
| MaskGCT [13] | ✗ | 6.382 | 0.717 | 0.645 | 0.23 $_{\pm0.20}$ | 4.24 $_{\pm0.16}$ | 4.38 $_{\pm0.14}$ |
| VALL-E [26] | ✓ | 10.721 | 0.403 | 0.485 | -1.04 $_{\pm0.50}$ | 3.12 $_{\pm0.41}$ | 2.77 $_{\pm0.45}$ |
| Voicebox [27] | ✗ | **6.181** | 0.475 | 0.575 | -0.55 $_{\pm0.22}$ | 3.93 $_{\pm0.25}$ | 3.49 $_{\pm0.29}$ |
| VoiceCraft [77] | ✓ | 10.072 | 0.438 | 0.517 | -0.39 $_{\pm0.22}$ | 3.51 $_{\pm0.33}$ | 3.29 $_{\pm0.28}$ |
| Vevo-TTS | ✓ | 9.673 | **0.544** | **0.579** | **0.12** $_{\pm0.20}$ | **4.11** $_{\pm0.20}$ | **4.12** $_{\pm0.21}$ |

| Model | AR? | WER (↓) | Speaker SIM (↑) | Emotion SIM (↑) | Naturalness CMOS (↑) | Speaker S-MOS (↑) | Emotion S-MOS (↑) |
|---|---|---|---|---|---|---|---|
| *EMOTION* | | | | | | | |
| Ground Truth | - | 11.792 | 0.673 | 0.936 | 0.00 | - | - |
| CosyVoice [24] | ✓ | 10.139 | 0.575 | 0.839 | -0.45 $_{\pm0.18}$ | 3.98 $_{\pm0.19}$ | 3.66 $_{\pm0.19}$ |
| MaskGCT [13] | ✗ | 12.502 | 0.600 | 0.822 | -0.31 $_{\pm0.17}$ | 4.07 $_{\pm0.16}$ | 3.76 $_{\pm0.25}$ |
| VALL-E [26] | ✓ | 15.731 | 0.396 | 0.735 | -1.43 $_{\pm0.33}$ | 2.52 $_{\pm0.38}$ | 2.63 $_{\pm0.36}$ |
| Voicebox [27] | ✗ | **12.647** | 0.451 | 0.811 | -0.65 $_{\pm0.20}$ | 3.81 $_{\pm0.16}$ | 3.61 $_{\pm0.19}$ |
| VoiceCraft [77] | ✓ | 16.042 | 0.345 | 0.788 | -0.60 $_{\pm0.24}$ | 3.42 $_{\pm0.31}$ | 3.52 $_{\pm0.25}$ |
| Vevo-TTS | ✓ | 14.458 | **0.466** | **0.840** | **-0.39** $_{\pm0.15}$ | **3.99** $_{\pm0.22}$ | **4.03** $_{\pm0.19}$ |

\* The best and the second best results among VALL-E, Voicebox, VoiceCraft, and Vevo-TTS are shown in **bold** and by underlined.

**Scoring criteria** 5 (Excellent, sounds like exactly the same accent), 4 (Good, sounds like a similar accent), 3 (Fair, sounds like a slightly similar accent), 2 (Poor, sounds like a different accent mostly), 1 (Bad, sounds like a completely different accent)

### E.2.4 EMOTION SIMILARITY MOS

**System Interface** One reference audio, One audio to be evaluated

**Questionnaire** Ignore the vocal characteristics (who is speaking), just pay attention to the emotion of the speaker. Is the emotion similar to the reference voice?

**Scoring criteria**   5 (Excellent, sounds like exactly the same emotion), 4 (Good, sounds like a similar emotion), 3 (Fair, sounds like a slightly similar emotion), 2 (Poor, sounds like a different emotion mostly), 1 (Bad, sounds like a completely different emotion)

### E.2.5   PROSODY SIMILARITY MOS

**System Interface**   One reference audio, One audio to be evaluated

**Questionnaire**   Ignore the vocal characteristics (who is speaking), just pay attention to the speaking style (how to speak). Is the speaking style (pace, tone, stress, intonation, pitch, emotion) consistent and identical with the reference voice?

**Scoring criteria**   5 (Excellent, sounds like a completely identical style), 4 (Good, sounds like a highly consistent style), 3 (Fair, sounds like a slightly similar style), 2 (Poor, sounds mostly like a different style), 1 (Bad, sounds like a completely different style)

### E.2.6   NATURALNESS CMOS

**System Interface**   One reference audio, One audio to be evaluated (with target text)

**Questionnaire**   Compared to the reference audio, is the quality and the human likeness of the audio to be evaluated better or worse?

**Scoring criteria**   -3 (Much worse), -2 (Worse), -1 (Slightly worse), 0 (No preference), 1 (Slightly better), 2 (Better), 3 (Much better)

### E.2.7   ACCENTEDNESS CMOS

**System Interface**   One accent label, One reference audio, One audio to be evaluated

**Questionnaire**   Assume that we want to generate the voice whose accent is *[accent label]*. Compared to the reference audio, is the accentedness of the audio to be evaluated better or worse?

**Scoring criteria**   -3 (Much worse), -2 (Worse), -1 (Slightly worse), 0 (No preference), 1 (Slightly better), 2 (Better), 3 (Much better)

### E.2.8   EMOTIVENESS CMOS

**System Interface**   One emotion label, One reference audio, One audio to be evaluated

**Questionnaire**   Assume that we want to generate the voice whose emotion is *[emotion label]*. Compared to the reference audio, is the emotional expressiveness of the audio to be evaluated better or worse?

**Scoring criteria**   -3 (Much worse), -2 (Worse), -1 (Slightly worse), 0 (No preference), 1 (Slightly better), 2 (Better), 3 (Much better)

## F   ETHICS STATEMENT

As with other powerful new AI innovations, we recognize this technology brings the potential for misuse and unintended harm. We will build a highly effective classifier that can distinguish between authentic speech and audio generated with Vevo to mitigate these possible future risks.

