# OpenReview forum: "Vevo: Controllable Zero-Shot Voice Imitation with Self-Supervised Disentanglement"
_ICLR.cc/2025/Conference — ICLR 2025 Poster_

### Official Review · Reviewer_PTWX · 2024-10-24

**Soundness:** 3
**Presentation:** 3
**Contribution:** 2
**Rating:** 6
**Confidence:** 4

**Summary:**

The paper addresses challenges in voice imitation for speech generation, proposing Vevo—a zero-shot framework with controllable timbre and style. It operates in two stages and uses a self-supervised dicoupling approach with VQ-VAE tokenizer. Trained on 60K hours of audiobook data, Vevo outperforms existing methods and shows effectiveness in zero-shot conversion and text-to-speech, demonstrating generalization and versatility.

**Strengths:**

1. Vevo can do both VC and TTS tasks and can simultaneously control both timbre and style.
2. The experiments are very detailed and the conclusions prove the effectiveness of the proposed model.
3. This paper is well structured and easy to read, and the demo sounds good.

**Weaknesses:**

Although the paper has done a lot of research on the comparison and selection of representations, it seems that there is relatively little innovati

**Questions:**

1. The content and content-style tokenizers are trained only on 100-hour data. Is the amount of data sufficient? Has a comparison of different amounts of data been made?
2. In section 4.1, "When K reduces further to 32, in addition to timbre, most style information is also filtered out", FPC drops from 0.764 to 0.706, that's not a big difference. How can it be proved that the style has been removed?

---

> ### Author Response · Authors · 2024-11-19
> **Response by Authors**
>
> We sincerely thank you for your appreciation of our experiments, results, demo, and paper writing. We will address your concerns as follows:
>
> ```
> Although the paper has done a lot of research on the comparison and selection of representations, it seems that there is relatively little innovation.
> ```
>
> We consider that our proposed self-supervised tokenizer offers several notable advantages. Compared to the existing supervised tokenizers, (1) It has a **simpler VQ design** with only one codebook and low bandwidth requirements (content/content-style tokens: 0.25/0.6 kbps, see details [in our common reply above](https://openreview.net/forum?id=anQDiQZhDP&noteId=OvBCMTxJ97)), (2) It is **independent from annotated corpora**, and (3) It has a **simpler and more stable training process**. These characteristics make it particularly suitable for scaling up in the current large-data era. We believe this is the main novelty of our proposed tokenizer.
>
> In addition to the tokenizer design, Vevo is able to achieve style conversion (such as accent and emotion conversion) in a zero-shot manner, without using any accented/emotional specific corpus or accent/emotion labels. We consider this to be another innovation of our work.
>
> ```
> The content and content-style tokenizers are trained only on 100-hour data. Is the amount of data sufficient? Has a comparison of different amounts of data been made?
> ```
>
> Thank you for your insightful question. [In our common reply above](https://openreview.net/forum?id=anQDiQZhDP&noteId=OvBCMTxJ97), we provided a detailed analysis of how varying training data scales affect tokenizer effectiveness. Specifically, we found that when training the tokenizer on English-only audiobook data, a 100-hour dataset was sufficient to achieve an effective tokenizer, with minimal gains from further data scaling. We hypothesize this is due to the relatively simple distribution of such data. Furthermore, in our recent extension work, we discovered that for in-the-wild and multilingual data, which has a more diverse distribution, increasing the tokenizer's training data scale is quite beneficial. Detailed results can be also found in the common reply above.
>
> ```
> In section 4.1, "When K reduces further to 32, in addition to timbre, most style information is also filtered out", FPC drops from 0.764 to 0.706, that's not a big difference. How can it be proved that the style has been removed?
> ```
>
> In Table 2, as we move from hidden features -> content-style tokens (K = 4096) -> content tokens (K = 32), the FPC values change from 0.824 -> 0.797 -> 0.706. Compared to content-style tokens and hidden features, we can see a obvious reduction in style information within content tokens (about 10% decrease). Beyond reducing K, we also employ duration reduction for content tokens to further remove style information (such as unit-level duration) from the token sequence. This approach has been validated by some speech-to-speech translation studies for its effectiveness in removing accent information [40].
>
> From the experimental results, Vevo has already demonstrated effective style imitation such as accent and emotion (Table 3, 4, and 5). We believe this is because under our current design, the style information gap between content-style tokens and content tokens is sufficient, which is enough to encourage the autoregressive transformer to imitate the content-style tokens from the style reference during continued generation.
>
> > [40] Textless speech-to-speech translation on real data. NAACL 2022.

---

> ### Author Response · Authors · 2024-11-25
> **Official Comment by Authors**
>
> Thanks again for your valuable comments, we would be grateful if we could hear your feedback regarding our answers. We would be happy to answer and discuss if you have further comments.

---

> ### Author Response · Authors · 2024-11-30
> **Looking forward to your reply**
>
> Dear reviewer:
>
> Thanks again for the time and effort you have dedicated to reviewing our paper! We greatly appreciate your insightful comments and suggestions. We have provided new experiments to address your concern, and we would love to receive feedback from you. If your concern is addressed, we humbly invite the reviewer to consider increasing the score. Your support means a lot to us!

---

### Official Review · Reviewer_z29c · 2024-11-02

**Soundness:** 4
**Presentation:** 3
**Contribution:** 3
**Rating:** 8
**Confidence:** 3

**Summary:**

This paper proposes a voice imitation approach supporting style information transfer at different granularities, including global timbre, accent, and emotion. The authors create varying sizes of information bottlenecks by VQ-VAE with different codebook sizes on self-supervised speech representations, obtaining speech representations with different levels of information. Upon these representations, the authors build a two-stage speech generation model composed of autoregressive and flow matching modules. Experiments demonstrate the effectiveness of this approach in voice imitation.

**Strengths:**

1. The idea of using codebooks of varying sizes to control the width of the information bottleneck is impressive, and the study on the order that information are filtered out with the shrinking codebook sizes are highly meaningful.

2. The experimental results are promising, and the demo sounds impressive.

**Weaknesses:**

I do not see any significant flaws in the paper for now. I may raise new questions based on the feedback from other reviewers.

**Questions:**

Did the authors validated their theory regarding the relationship between VQ-VAE codebook size and the information contained in representations on other self-supervised speech representations (such as wav2vec2, w2vbert)? Are their conclusions consistent?

---

> ### Author Response · Authors · 2024-11-19
> **Response by Authors**
>
> We sincerely thank you for your positive feedback regarding our idea, experiments, results, and demo.
>
> ```
> Did the authors validate their theory regarding the relationship between VQ-VAE codebook size and the information contained in representations of other self-supervised speech representations (such as wav2vec2, w2vbert)? Are their conclusions consistent?
> ```
>
> Thank you for raising such an insightful question. In our recent ongoing work, we have conducted similar explorations with Whisper [71]. Specifically, we extract Whisper's encoder output as hidden features, which can be considered as intermediate representations between speech (its input) and text (its output), containing timbre, content, and style information. During the VQ-VAE quantization process of these features, as we reduce the codebook's vocabulary size, we observe that information is filtered out in the order of timbre -> style -> content, which aligns with Vevo's findings.
>
> In some concurrent works within the zero-shot TTS field, such as MaskGCT [13], similar VQ-VAE-based quantization has been performed on [w2vbert-2.0](https://huggingface.co/facebook/w2v-bert-2.0). They employed a vocabulary size of 8192 for content-style tokens (which they termed "semantic tokens"), whose codebook size is two times larger than our chosen 4096. While these tokens demonstrate excellent performance in TTS tasks, they exhibit some timbre leakage issues when directly applied to voice conversion tasks [13], which indicates that too large vocabulary sizes could lead to residual timbre information. This phenomenon corroborates our findings.
>
> > [13] Maskgct: Zero-shot text-to-speech with masked generative codec transformer. arXiv 2024.
> >
> > [71] Robust speech recognition via large-scale weak supervision. ICML 2023.

---

### Official Review · Reviewer_1b8h · 2024-11-03

**Soundness:** 2
**Presentation:** 2
**Contribution:** 2
**Rating:** 3
**Confidence:** 4

**Summary:**

This paper builds upon the foundational HuBERT representation model and explores the disentanglement of content tokens and content-style tokens by analyzing VQ codebook spaces of varying sizes. Based on this disentanglement, the authors propose a cascaded generation framework called Vevo (AR Transformer combined with Flow Matching), which successfully performs zero-shot timbre, style, and voice imitation tasks.

**Strengths:**

1. Exploring the unified approach to zero-shot timbre, style, and voice imitation tasks through disentangled representations is a meaningful attempt, although similar efforts have been made previously.
2. The novel approach of disentangling timbre, style, and content by analyzing different sizes of VQ codebook spaces is interesting, although it is difficult to provide strict evidence of its effectiveness.

**Weaknesses:**

1. It is challenging to guarantee strictly disentangled results when analyzing different sizes of VQ codebook spaces for timbre, style, and content.

&nbsp; &nbsp; 1.1. The results may be biased by hyperparameter tuning in downstream tasks.

&nbsp; &nbsp; 1.2. The authors should include comparisons with other disentanglement methods in Table 2, such as label-based    disentanglement techniques.

&nbsp; &nbsp; 1.3. The differences in evaluation metrics between various codebook sizes, such as a SIM score improvement from 40 to 44, are minimal and insufficient to claim successful disentanglement.

&nbsp; &nbsp; 1.4. The performance metrics in Table 2 for different codebook sizes show substantial gaps compared to the ground truth (GT).

2. The overall cascaded generation framework of Vevo is not highly innovative, as similar frameworks have already been proposed, such as SeedTTS and CosyVoice.

3. The unified definition of zero-shot timbre, style, and voice imitation tasks is not new, as seen in prior works like ControlSpeech and StyleFusion TTS.

4. The experimental results do not demonstrate significant performance gains. Furthermore, the baselines are relatively weak. At least, comparisons with open-source models like CosyVoice should be included. Additionally, comparing to FaCodec is unfair; a comparison with NaturalSpeech3 would be more appropriate

**Questions:**

1. I am unclear about the specific implementation details of the Vevo-Timbre and Vevo-Style versions.

2. Will you be releasing the complete training and inference code, as well as the model weights?

---

> ### Author Response · Authors · 2024-11-19
> **Response by Authors (Part 1/3)**
>
> Thank you for your insightful comments and we sincerely appreciate your recognition of our work as "a meaningful attempt" and "a novel approach". We will address your concerns as follows:
>
> ```
> 1. It is challenging to guarantee strictly disentangled results when analyzing different sizes of VQ codebook spaces for timbre, style, and content.
>      1.1. The results may be biased by hyperparameter tuning in downstream tasks.
>      1.2. The authors should include comparisons with other disentanglement methods in Table 2, such as label-based disentanglement techniques.
> ```
>
> We kindly clarify that **we have conducted comparisons for several disentanglement techniques**, including **ASR tokens** (label-based; Table 2 and Table 4, serving as a type of content tokens), **FACodec** (label-based; Table 3, serving as a type of content-style tokens), and **K-means tokens** (Table 2, a widely used method for disentanglement). [In our common reply above](https://openreview.net/forum?id=anQDiQZhDP&noteId=OvBCMTxJ97), we provided a detailed analysis comparing our proposed self-supervised tokenizer with existing label-based tokenizers, such as FACodec, focusing on architectural design trade-offs and the effectiveness of our method in zero-shot timbre imitation tasks.
>
> We acknowledge that our proposed self-supervised tokenizer requires careful parameter tuning. However, it offers several notable advantages that the existing supervised tokenizers (like FACodec) do not have: (1) **simpler VQ design** with only one codebook and low bandwidth requirements (content/content-style tokens: 0.25/0.6 kbps), (2) **independence from annotated corpora**, and (3) **simpler and more stable training process**. These characteristics make it particularly suitable for scaling up in the current large-data era. While theoretically our current design may not be the optimal choice, our parameter configuration has already demonstrated promising results across various zero-shot voice imitation tasks. We plan to investigate the integration of self-supervised and supervised tokenizers in future work, aiming to potentially achieve both superior performance and enhanced interpretability (i.e., injecting prior domain knowledge from supervised learning).
>
> ```
> 1.3. The differences in evaluation metrics between various codebook sizes, such as a SIM score improvement from 40 to 44, are minimal and insufficient to claim successful disentanglement.
> ```
>
> We apologize but we are unable to identify the specific "40 to 44" comparison you mentioned. In Table 2, if we highlight the three most critical rows for comparison among (1) 18th layer hidden features (i.e., the starting point for our disentanglement), (2) VQ-VAE tokens #Vocab=4096 (i.e., our proposed content-style tokens), and (3) VQ-VAE tokens #Vocab=32 (i.e., our proposed content tokens):
>
> | Representations               | WER $\downarrow$ | Speaker-SIM (to ref) $\uparrow$ | Speaker-SIM (to src) $\downarrow$ | FPC (to src) $\uparrow$ |
> | -------------------- | ---------------- | ---------------------------------- | ------------------------------------ | -------------------------- |
> | Hidden features      | 5.324            | 0.250                              | 0.505                                | 0.824                      |
> | Content-style Tokens | 6.908            | 0.403                              | 0.236                                | 0.797                      |
> | Content Tokens       | 9.731            | 0.426                              | 0.161                                | 0.706                      |
>
> We can observe that:
> 1. **From hidden features to content-style tokens**, we can see that **the timbre leakage issue is significantly mitigated**: Speaker-SIM (to ref) increases from 0.250 to 0.403 (absolute/relative change: 15.3% / 61.2%), Speaker-SIM (to src) decreases from 0.505 to 0.236 (absolute/relative change: 26.9% / 53.3%). We consider these to be significant differences. Meanwhile, style information is largely preserved (FPC: 0.824 -> 0.797).
> 2. **From content-style tokens to content tokens**, we observe **substantial removal of style information**: FPC decreases from 0.797 to 0.706 (absolute/relative change: 9.1% / 11.4%). We consider this to be a notable difference in style information retention.
>
> ```
> 1.4. The performance metrics in Table 2 for different codebook sizes show substantial gaps compared to the ground truth (GT).
> ```
>
> Yes, not only Vevo, but also almost all existing methods show substantial performance gaps compared to the ground truth (see Tables 2, 3, and 5). **This is just a strong evidence that our research problem is challenging and our efforts are meaningful.** Although Vevo has matched or surpassed existing methods in zero-shot timbre, voice, and style conversion tasks, the controllable zero-shot voice imitation problem still merits continued exploration by the broader research community.

---

> ### Author Response · Authors · 2024-11-19
> **Response by Authors (Part 2/3)**
>
> ```
> 2. The overall cascaded generation framework of Vevo is not highly innovative, as similar frameworks have already been proposed, such as SeedTTS and CosyVoice.
> ```
>
> In the zero-shot TTS field, several works have indeed adopted a cascaded generation framework, such as TorToise-TTS [25], SeedTTS [11], CosyVoice [24], and MaskGCT [13]. However, **Vevo extends beyond a zero-shot TTS system to become a unified and versatile voice imitation framework**. In addition to zero-shot TTS, Vevo can also perform more tasks such as zero-shot voice conversion and zero-shot accent/emotion conversion.
>
> Compared to existing TTS works, Vevo's novelty lies in two key aspects:
>
> 1. **Our proposed content-style tokens successfully decouple the timbre information, enabling timbre-controllable generation during the acoustic modeling stage.** In contrast, existing TTS models' content-style tokens (sometimes referred to as "semantic tokens") suffer from timbre leakage issues. Consequently, to address voice conversion, these models require an additional fine-tuning stage (such as the "self-distillation fine-tuning stage" in SeedTTS [11] and "timbre perturbation based fine-tuning stage" in MaskGCT [13]).
>
> 2. **Beyond text input, Vevo can also process speech input. This capability enables zero-shot accent/emotion conversion** — given a speech input, Vevo can modify its accent/emotion while preserving the speaker identity, which can be prompted through a style reference. We consider this a significant contribution, as existing works in accent/emotion conversion areas rarely achieve zero-shot conversion. Notably, Vevo accomplishes this without requiring any accented/emotional specific corpus or accent/emotion labels.
>
> ```
> 3. The unified definition of zero-shot timbre, style, and voice imitation tasks is not new, as seen in prior works like ControlSpeech and StyleFusion TTS.
> ```
>
> We kindly clarify that our defined zero-shot timbre imitation, style imitation, and voice imitation (conversion tasks) all **use wav rather than text as input** (see Table 1), **which significantly differentiates them from TTS tasks** (such as ControlSpeech and StyleFusion TTS). The key distinction between Vevo and ControlSpeech lies in their control mechanisms: ControlSpeech employs textual descriptions (natural language) for style control, while Vevo uses a style reference (speech) as the imitation target. We believe both approaches are meaningful for real-world applications and warrant further investigation.
>
> We would like to point out that StyleFusion TTS was released at [arXiv](https://arxiv.org/abs/2409.15741) on September 24, 2024, just one week before the ICLR submission deadline (October 1, 2024). **We believe it would be unfair to draw comparisons between Vevo and StyleFusion TTS.**
>
> ```
> 4. The experimental results do not demonstrate significant performance gains. Furthermore, the baselines are relatively weak. At least, comparisons with open-source models like CosyVoice should be included. Additionally, comparing to FaCodec is unfair; a comparison with NaturalSpeech3 would be more appropriate.
> ```
>
> **We kindly clarify that we have compared with CosyVoice** (in Table 5), and our [demo page](https://versavoice.github.io/#vevo-tts) includes comparison samples between CosyVoice and Vevo-TTS. As shown in Table 5, despite being trained on only 60K hours of audiobook data, Vevo outperforms CosyVoice (that is trained on 171K hours of in-the-wild data) in subjective evaluations including Naturalness-CMOS (N-CMOS), Accent Similarity MOS (AS-MOS), and Emotion Similarity MOS (ES-MOS).
>
> We have made our best effort to include strong baselines from recent top-tier venues and high-quality public implementations across various imitation tasks. Specifically, our baseline selection includes:
>
> 1. For zero-shot timbre and voice imitation tasks (Table 3):
>    - UniAudio (ICML 2024)
>    - FACodec (ICML 2024)
>
> 2. For zero-shot style imitation tasks (Table 4):
>    - Conv-Speak (MM 2024)
>    - VoiceShop (2024, authored by SeedTTS's Team)
>
> 3. For zero-shot TTS tasks (Table 5):
>    - VoiceCraft (ACL 2024)
>    - Voicebox (NeurIPS 2023)
>    - CosyVoice (2024)
>    - MaskGCT (2024)
>
> Regarding NaturalSpeech3, as it is not open-sourced, we were unable to evaluate it on our ACCENT and EMOTION sets (Table 5), where accent and emotion are two of the most representative attributes for style-imitated generation.

---

> ### Author Response · Authors · 2024-11-19
> **Response by Authors (Part 3/3)**
>
> ```
> Implementation details of Vevo-Timbre and Vevo-Style.
> ```
>
> As described in Section 3.4, Vevo-Timbre only uses $M_{acoustic}$. Vevo-Style requires both $M_{style}$ and $M_{acoustic}$, where we use the source speech as the timbre reference for $M_{acoustic}$ to ensure that speaker identity is preserved during style conversion. The detailed implementation specifications for both $M_{style}$ and $M_{acoustic}$ can be found in Appendix B.2 and B.4, respectively.
>
> ```
> Will you be releasing the complete training and inference code, as well as the model weights?
> ```
>
> Yes, we plan to release all the code along with the pre-trained models. We are currently working on extension experiments training Vevo on larger-scale in-the-wild and multilingual datasets. We will include these latest pre-trained models in our open-source release.
>
> > [11] Seed-tts: A family of high-quality versatile speech generation models. arXiv 2024.
> >
> > [13] Maskgct: Zero-shot text-to-speech with masked generative codec transformer. arXiv 2024.
> >
> > [24] Cosyvoice: A scalable multilingual zero-shot text-to-speech synthesizer based on supervised semantic tokens. arXiv 2024.
> >
> > [25] Better speech synthesis through scaling. arXiv 2023.

---

> ### Author Response · Authors · 2024-11-25
> **Official Comment by Authors**
>
> Thanks again for your valuable comments, we would be grateful if we could hear your feedback regarding our answers. We would be happy to answer and discuss if you have further comments.

---

> > ### Comment · Reviewer_1b8h · 2024-11-26
> >
> > I appreciate the author's responses to my questions, which have addressed some of my concerns. However, I still have reservations regarding certain aspects of the experimental design and methodology, specifically the contributions related to Weaknesses 1, 2, 3, and 4.
> >
> > 1. The core decoupling proposed by the author relies on the design of an information bottleneck via vocabulary size, with the claim that "As the vocabulary size decreases, timbre imitation improves." However, the results in Table 2, which show an ablation from 0.398 to 0.441, do not seem to reflect a significant improvement. Perhaps I have misunderstood some aspect, and I believe this point warrants further clarification.
> >
> > 2. I also do not observe a significant issue of timbre leakage in cascaded structures, such as the semantic tokens in CosyVoice, as suggested by the author.
> >
> > 3. The key innovation of this work lies in the independent control of style, timbre, and content. As such, I do not fully agree with the author's distinction between inputs, as I believe this aspect has been addressed in prior work. Furthermore, it appears that ControlSpeech may support speech-controlled style (I am not entirely certain), since the ground truth style is extracted directly from the speech during training.
> >
> > 4. From the results in Table 5, I observe that the performance of Vevo is significantly lower than that of CosyVoice. I do not entirely agree with the author's rebuttal, which reports only the positive results while omitting the less favorable outcomes.
> >
> > Finally, I am acutely aware of the rigorous standards upheld by ICLR with regard to methodological innovation and experimental results. While I acknowledge some of the contributions of Vevo, given the overall standards of ICLR, I have decided to lower my score from 5 to 3.

---

> ### Author Response · Authors · 2024-11-30
> **Second-round Response by Authors (Part 1/4)**
>
> Thanks for your feedback. We will address your concerns as follows:
>
> ```
> 1. The core decoupling proposed by the author relies on the design of an information bottleneck via vocabulary size, with the claim that "As the vocabulary size decreases, timbre imitation improves." However, the results in Table 2, which show an ablation from 0.398 to 0.441, do not seem to reflect a significant improvement. Perhaps I have misunderstood some aspect, and I believe this point warrants further clarification.
> ```
>
> - Regarding the timbre imitation (or timbre leakage) issues, we suppose that **both S-SIM (to ref) and  S-SIM (to src) should be considered together**. As shown in Table 2, when the vocabulary size is reduced from 16384 to 16, the S-SIM (to ref) increases from 0.398 to 0.441, as you noted. In addition, **the S-SIM (to src), which significantly decreases from 0.306 to 0.146, should not be overlooked**. This shift indicates a substantial reduction in timbre leakage from the source, which is an important factor in evaluating the performance of timbre imitation.
>
> - Even when considering only S-SIM (to ref), as we reduce the vocabulary size from infinity (i.e., 18th hidden features) to 16384, we observe a significant increase in S-SIM (to ref) from 0.250 to 0.398. This suggests that even a relatively large vocabulary size of 16384 can help mitigate the timbre leakage issue. However, this vocabulary size is still insufficient to fully address it, as indicated by the remaining considerable S-SIM (to src) of 0.306. Therefore, we further reduce the vocabulary size to 4096, with S-SIM (to src) further decreasing to 0.236, to obtain our proposed content-style tokens.
>
> ```
> 2. I also do not observe a significant issue of timbre leakage in cascaded structures, such as the semantic tokens in CosyVoice, as suggested by the author.
> ```
>
> - **We have never claimed that CosyVoice [24] has timbre leakage issues**. Our original statements in the submitted paper and during the first-round response are as follows:
>     - In the main text of our paper (Page 2, L68), our original statement is: *“Secondly, the decoupling of timbre and style in existing methods is still insufficient, making it challenging to control them independently, unless mitigated by some timbre (or style) perturbations or additional fine-tuning stages [11, 13, 33].”* In this context, **SeedTTS [11] and MaskGCT [13] are referenced as examples of the timbre perturbation issue, as two competitive models in the recent TTS field**.
>     - During the first-round response, our original statement is: *“In contrast, existing TTS models' content-style tokens (sometimes referred to as "semantic tokens") suffer from timbre leakage issues. Consequently, to address voice conversion, these models require an additional fine-tuning stage (such as the 'self-distillation fine-tuning stage' in SeedTTS [11] and 'timbre perturbation based fine-tuning stage' in MaskGCT [13]).”* We apologize if the phrasing "existing TTS models" appeared too absolute; what we intended to convey was “some existing TTS models.” Nevertheless, **the references we provided for evidence are still SeedTTS [11] and MaskGCT [13].**
>
> - Regarding the timbre leakage issues of the representations in SeedTTS and MaskGCT, the specific references in their original texts are as follows:
>     - SeedTTS [11]: See Section 4.1 and Table 6. Specifically, in Table 6 of their paper, it can be observed that after the self-distillation fine-tuning stage, the Speaker SIM on English evaluation set **increases from 0.491 to 0.753**.
>     - MaskGCT [13]: See Appendix I, which states: *"The zero-shot voice conversion task aims to alter the source speech to sound like that of a target speaker using a reference speech from the target speaker, without changing the semantic content. We can directly use the semantic tokens $S_{src}$ extracted from the source speech and the prompt acoustic tokens $A_{ref}$ extracted from the reference speech to predict the target acoustic tokens $A_{tgt}$. Since **$S_{src}$ may retain some timbre information**, we perform timbral perturbation on the semantic features input to the semantic codec encoder."*
>
> > [11] Seed-tts: A family of high-quality versatile speech generation models. arXiv 2024.
> >
> > [13] Maskgct: Zero-shot text-to-speech with masked generative codec transformer. arXiv 2024.
> >
> > [24] Cosyvoice: A scalable multilingual zero-shot text-to-speech synthesizer based on supervised semantic tokens. arXiv 2024.

---

> ### Author Response · Authors · 2024-11-30
> **Second-round Response by Authors (Part 2/4)**
>
> ```
> 3. The key innovation of this work lies in the independent control of style, timbre, and content. As such, I do not fully agree with the author's distinction between inputs, as I believe this aspect has been addressed in prior work. Furthermore, it appears that ControlSpeech may support speech-controlled style (I am not entirely certain), since the ground truth style is extracted directly from the speech during training.
> ```
>
> 1. We are somewhat confused by the phrase *"the authors' distinction between inputs"*. Based on our first-round response above, we conjecture (please correct us if we are wrong) that the reviewer intends to suggest our claim *"Beyond text input, Vevo can also process speech input. This capability enables zero-shot accent/emotion conversion"* not be convincing. However, we respectfully disagree with this point. On the contrary, **we believe that the ability of Vevo to accept both text and speech as input is important**. The reasons are as follows:
>     - Firstly, if speech cannot be used as input, how could we conduct all the **conversion** tasks (e.g., voice conversion, emotion conversion, accent conversion) discussed in our paper? Of course, from an engineering perspective, one could use an additional ASR model to first recognize the speech and convert it to text, and then apply TTS technology to perform these conversion tasks (i.e., turning conversion tasks into ASR + TTS). However, this solution is subject to the biases introduced by the ASR stage, such as ASR recognition errors and latency issues. Therefore, from a scientific research perspective, we believe it is meaningful to study using speech as input directly. This approach is more straightforward and better aligns with the end-to-end vision of deep learning. **We argue that this is also the value of the entire "conversion" field, including voice conversion, emotion conversion, accent conversion, and more.**
>     - Secondly, when controlling style and timbre, **the required capabilities of the model differ depending on whether the input is speech or text.** For example, in a zero-shot voice imitation task, given a reference, when the input is text, we only need to add the reference's style and timbre information to the content information (i.e., the input text), which follows a typical zero-shot TTS paradigm. However, when the input is speech, we first need to remove the style and timbre information from the input speech to extract the content information (i.e., our proposed content tokens), and then add the reference's style and timbre information to this content. **In other words, the model needs to implicitly learn both ASR and TTS capabilities simultaneously when taking speech as input.**
>
> 2. We are somewhat unclear about what *"this aspect"* refers to in the reviewer's comment *"as I believe this aspect has been addressed in prior work."* Based on the reviewer's comments above, we conjecture (please correct us if we are wrong) that the reviewer is suggesting *the independent control of style, timbre, and content has been addressed in prior work*. Additionally, we assume that by *"prior work"*, the reviewer is referring to ControlSpeech [91] and StyleFusion TTS [92]. **We would like to respectfully emphasize that this argument is NOT true**, and we elaborate as follows:
>     - Regarding ControlSpeech [91] (where you mention being *"not entirely certain"*), we would like to point out that, on the one hand, ControlSpeech does not conduct any experiments where **speech is directly used as a style reference**, which is a unique aspect of Vevo. On the other hand, **ControlSpeech specifically acknowledges that extracting a precise textual style description from speech is highly non-trivial**. This is why the authors of ControlSpeech have put significant effort into creating a speech dataset with textual style descriptions (i.e., VccmDataset), which represents a great contribution to the task of **natural language-controlled** TTS. We want to emphasize again that we believe both approaches—using natural speech language or, as in Vevo, directly using speech as a style reference—are meaningful for real-world applications and warrant further exploration.
>     - Regarding StyleFusion TTS [92], first, we have pointed out in our first-round response that StyleFusion TTS was published just one week before the ICLR submission deadline, making it unfair to draw direct comparisons. Second, StyleFusion also only accepts text as input and does not perform tasks like voice conversion, accent conversion, or emotion conversion, which are part of the broader range of conversions that Vevo can handle.
>
> > [91] ControlSpeech: Towards Simultaneous Zero-shot Speaker Cloning and Zero-shot Language Style Control With Decoupled Codec. arXiv 2024.
> >
> > [92] StyleFusion TTS: Multimodal Style-control and Enhanced Feature Fusion for Zero-shot Text-to-speech Synthesis. arXiv 2024.
>
> **(... See Part 3 ...)**
>
> 3. Finally, we would like to highlight...

---

> ### Author Response · Authors · 2024-11-30
> **Second-round Response by Authors (Part 3/4)**
>
> **(... Continued ...)**
>
> 3. Finally, we would like to highlight two unique aspects of our work:
>     - The ability to perform **both style-preserved voice conversion** (i.e., timbre imitation in our paper, by Vevo-Timbre) **and style-converted voice conversion** (i.e., voice imitation in our paper, by Vevo-Voice).
>     - The ability to perform **zero-shot accent/emotion conversion** (i.e., style imitation in our paper, by Vevo-Style).
>
> **Could you provide specific prior works that are capable of addressing such tasks?** We would be happy to include them as baselines and further improve our research.
>
> ```
> 4. From the results in Table 5, I observe that the performance of Vevo is significantly lower than that of CosyVoice. I do not entirely agree with the author's rebuttal, which reports only the positive results while omitting the less favorable outcomes.
> ```
>
> Before providing our response to this comment, we would like to point out that:
>
> - CosyVoice was [released on July 7 at arXiv](https://arxiv.org/abs/2407.05407). **According to the guidelines of ICLR, CosyVoice and Vevo are concurrent work, and thus, CosyVoice cannot serve as a basis for comparison in the review process.** Additionally, in terms of training data, CosyVoice uses 171K hours of in-the-wild multilingual data, whereas Vevo-TTS uses only 60K hours of English-only audiobook data (as annotated in column 3 of Table 5). Directly comparing CosyVoice and Vevo-TTS would therefore be unfair.
> > Q: What constitutes concurrent/contemporaneous work, and what is the relevant policy regarding it?
> >
> > A: We consider papers contemporaneous if they are published within the last four months. That means, since our full paper deadline is October 1, **if a paper was published (i.e., at a peer-reviewed venue) on or after July 1, 2024, authors are not required to compare their own work to that paper.** Authors are encouraged to cite and discuss all relevant papers, but they may be excused for not knowing about papers not published in peer-reviewed conference proceedings or journals, which includes papers exclusively available on arXiv.
> >
> >  -- https://iclr.cc/Conferences/2025/FAQ
>
> - However, to highlight the difference in performance between Vevo-TTS and the latest state-of-the-art TTS models, we chose to include a comparison with CosyVoice in the submitted paper. We have clearly presented all the results and have not "omitted" anything.
> - Furthermore, the results in Table 5 show that **Vevo-TTS even outperforms CosyVoice in three subjective evaluation metrics** (including the N-CMOS, AS-MOS, and ES-MOS), which has been mentioned in our first-round response. **We believe that these results strengthen our paper, rather than detracting from it.**
>
> Towards the reviewer comment, our response is as follows:
> - Firstly, we suppose that **the statement "the performance of Vevo is significantly lower than that of CosyVoice" is NOT true**, for the following reasons:
>     - In the field of speech generation, it is well-known that subjective evaluations better align with human auditory perception than objective evaluations. As shown in Table 5, for the four **subjective** metrics, Vevo-TTS outperforms CosyVoice in N-CMOS, AS-MOS, and ES-MOS, with the exception of SS-MOS, where Vevo-TTS is slightly lower than CosyVoice (4.05 vs. 4.11).
>     - We assume that the reviewer’s concerns are primarily focused on **objective** evaluation results. Specifically, in Table 5, Vevo-TTS matches CosyVoice in E-SIM, but falls short in WER, S-SIM, and A-SIM. We hypothesize that this discrepancy arises from CosyVoice benefiting from its larger-scale training data (171K hours, about three times the size of Vevo-TTS) and its in-the-wild training setting. As shown in Table 5, models trained on such large-scale, in-the-wild data, like MaskGCT and CosyVoice, significantly outperform models like VALL-E, Voicebox, VoiceCraft, and Vevo-TTS in S-SIM and A-SIM, with MaskGCT and CosyVoice even surpassing the ground truth in A-SIM. We would like to include this discussion in our revised paper to provide a clearer context for the performance differences.
> - Besides, **we disagree with the claim that we "omit the less favorable outcomes." In Table 5 of our submitted paper, we have clearly presented all the results, without hiding or omitting anything. During the rebuttal period, we have not modified or concealed any of these results.** In our first-round response, to address the reviewer concerns about "performance gains", we stated, *"Vevo outperforms CosyVoice (which is trained on 171K hours of in-the-wild data) in subjective evaluations including Naturalness-CMOS (N-CMOS), Accent Similarity MOS (AS-MOS), and Emotion Similarity MOS (ES-MOS)."* We believe this statement is both objective and accurate. Moreover, as we previously emphasized, subjective evaluation metrics are better aligned with human auditory perception, which is why we once again highlighted the subjective metrics results in our rebuttal.

---

> ### Author Response · Authors · 2024-11-30
> **Second-round Response by Authors (Part 4/4)**
>
> # Additional Results Compared to CosyVoice on Zero-Shot Timbre and Voice Imitation Tasks
>
> As we mentioned in the previous comment, CosyVoice and Vevo can be considered as concurrent works. Besides, In our submitted paper, we have already presented a comparison between Vevo and CosyVoice on the zero-shot TTS task. However, **we believe that the comparison to CosyVoice is one of the major concerns of the reviewer.** Therefore, to further verify the effectiveness of Vevo, we would like to introduce additional experimental results. Specifically, we will present:
> 1. The results of CosyVoice's Flow Matching model (referred to as **CosyVoice-Timbre**) on the zero-shot timbre imitation and voice imitation tasks.
> 2. The results of Vevo-Timbre and Vevo-Voice **trained on large-scale in-the-wild data**, which is similar to CosyVoice. Specifically, we use [Emilia](https://arxiv.org/abs/2407.05361) as the training data, which includes 100K hours of speech and covers six languages (English, Chinese, German, French, Japanese, and Korean).
>
> The experimental results are as follows:
>
> ## Zero-Shot Timbre Imitation Task
>
> *Note: We propose **Vevo-Timbre** to address this task in the paper. Vevo-Voice can be considered as a baseline here. The experimental settings are the same as Table 3.*
>
> | Model                           | Training Data                | WER $\downarrow$ | Speaker SIM $\uparrow$ | FPC $\uparrow$ |
> | ------------------------------- | ---------------------------- | ---------------- | ---------------------- | -------------- |
> | CosyVoice-Timbre                | 171K hours, In-the-wild data | 3.828            | 0.469                  | 0.608          |
> | Vevo-Voice *(reported in paper)*  | 60k hours, Audiobook data    | 7.694            | 0.458                  | 0.485          |
> | Vevo-Voice                      | 100K hours, In-the-wild data | 6.080            | **0.543**              | 0.547          |
> | Vevo-Timbre *(reported in paper)* | 60k hours, Audiobook data    | **2.968**        | 0.420                  | **0.686**      |
> | Vevo-Timbre                     | 100K hours, In-the-wild data | *3.531*          | *0.481*                | *0.685*        |
>
> From the table above, we can observe the following:
> - The Vevo-Timbre model reported in our paper (which is trained on 60K-hour audiobook data) performs better than CosyVoice-Timbre in terms of WER and FPC, while CosyVoice-Timbre outperforms Vevo-Timbre in Speaker SIM.
> - When trained on 100K-hour in-the-wild data, Vevo-Timbre surpasses CosyVoice-Timbre in all three metrics: WER, Speaker SIM, and FPC.These results also highlight the effectiveness of Vevo when trained on large-scale, in-the-wild data.
>
> ## Zero-Shot Voice Imitation Task
>
> *Note: We propose **Vevo-Voice** to address this task in the paper. Vevo-Timbre can be considered as a baseline here. The experimental settings are the same as Table 3.*
>
> | Model                           | Training Data                | WER $\downarrow$ | Speaker SIM $\uparrow$ | Accent SIM $\uparrow$ | Emotion SIM $\uparrow$ |
> | ------------------------------- | ---------------------------- | ---------------- | ---------------------- | --------------------- | ---------------------- |
> | CosyVoice-Timbre                | 171K hours, In-the-wild data | 12.449           | 0.601                  | 0.602                 | 0.711                  |
> | Vevo-Timbre *(reported in paper)* | 60k hours, Audiobook data    | *12.351*         | 0.486                  | 0.567                 | 0.816                  |
> | Vevo-Timbre                     | 100K hours, In-the-wild data | **12.191**       | *0.609*                | *0.630*               | *0.682*                |
> | Vevo-Voice *(reported in paper)*  | 60k hours, Audiobook data    | 15.214           | 0.517                  | 0.614                 | **0.872**              |
> | Vevo-Voice                      | 100K hours, In-the-wild data | 14.243           | **0.674**              | **0.733**             | *0.868*                |
>
> From the table above, we can observe the following:
> - The Vevo-Voice model reported in our paper (which is trained on 60K-hour audiobook data) performs better than CosyVoice-Timbre in style imitation, such as Accent/Emotion SIM. However, CosyVoice-Timbre outperforms Vevo-Voice in terms of WER and Speaker SIM.
> - When trained on 100K-hour in-the-wild data, Vevo-Voice performs worse than CosyVoice-Timbre in WER (due to the autoregressive stage used in Vevo-Voice). However, Vevo-Voice outperforms CosyVoice-Timbre in both Speaker SIM and Accent/Emotion SIM. These results demonstrate Vevo's strong performance in style imitation tasks, particularly when trained on large-scale, in-the-wild data.
>
> We will include these results, discussions, and all of the pre-trained models in our revised paper and future open-source release. If you have any further questions or concerns, please feel free to let us know.

---

### Official Review · Reviewer_EEAE · 2024-11-04

**Soundness:** 3
**Presentation:** 3
**Contribution:** 2
**Rating:** 5
**Confidence:** 3

**Summary:**

The paper introduces Vevo, a zero-shot voice imitation framework with controllable timbre and style, featuring a fully self-supervised approach that progressively decouples timbre, style, and linguistic content of speech using VQ-VAE tokenizers with an information bottleneck technique. Vevo outperforms existing models in accent and emotion conversion, particularly excelling in zero-shot tasks, demonstrating its versatility and robustness.

**Strengths:**

1. This paper introduces a unified speech synthesis framework, Vevo, capable of handling zero-shot timbre, style, and voice imitation tasks. It provides clear and precise definitions of timbre and style and achieves its objectives through a two-stage modeling strategy: Content-Style Modeling and Acoustic Modeling.

2. By experimenting with the codebook size of self-supervised speech representations quantized via VQ-VAE, the paper follows the theory of information bottleneck and implements a progressive information filtering approach. This results in a simple yet efficient self-supervised method to gradually decouple timbre, style, and linguistic content.

3. Experiments indicate that Vevo matches or exceeds the performance of comparable state-of-the-art speech generation models across various zero-shot scenarios, showcasing its strong generalization and versatility.

**Weaknesses:**

1. The self-supervised VQ-VAE tokenizer for content and content-style information requires meticulous parameter tuning, such as continually adjusting the codebook size. Balancing the performance between the content and content-style tokenizers involves trade-offs, making it a suboptimal choice with potential risks. It might be beneficial to compare this with supervised tokenizer models, such as extracting content and content-style tokens directly using open-source models like FACodec \[1], to evaluate the impact on model performance and make more informed decisions.

2. In the Content-Style to Acoustic stage, the paper utilizes a conditional flow matching model with a masking strategy, relying on unmasked portions of mel spectrograms to guide the modeling of masked timbre, linguistic content, and acoustic style. This stage may encounter issues of information leakage, where the model might learn the speaker's context from the provided unmasked mel portions during training, aiding in better generation of masked mel segments. During inference, while aiming to replicate the timbre reference, the model might inadvertently also learn consistent style information.

\[1]: Ju Z, Wang Y, Shen K, et al. Naturalspeech 3: Zero-shot speech synthesis with factorized codec and diffusion models[J]. arXiv preprint arXiv:2403.03100, 2024.

**Questions:**

1. The paper notes that "both the content and content-style tokenizers are trained on a 100-hour subset randomly sampled from the full 60K-hour dataset." Training on such a small subset raises questions about the model's generalization and robustness. How are these ensured given the limited training data?

2. Has the model been tested on datasets beyond audiobooks to evaluate its zero-shot voice generation capabilities, including timbre imitation, style imitation, and voice imitation?

3. Regarding the parameters in the conditional flow matching model, how is the "randomly mask ratio" determined, especially given that the paper reports a range of "70%-100%"?

---

> ### Author Response · Authors · 2024-11-19
> **Reponse by Authors (Part 1/2)**
>
> Thank you for your insightful comments and we really appreciate your consideration of our method "simple yet efficient". We will address your concerns as follows:
>
> ```
> About the proposed self-supervised tokenizer: Require meticulous parameter tuning and is a suboptimal choice; Need to compare with supervised tokenizers like FACodec.
> ```
>
> We kindly clarify that **we have conducted comparisons for several supervised tokenizers**, including **ASR tokens** (Table 2 and Table 4, serving as a type of content tokens) **and FACodec** (Table 3, serving as a type of content-style tokens). [In our common reply above](https://openreview.net/forum?id=anQDiQZhDP&noteId=OvBCMTxJ97), we provided a detailed analysis comparing our proposed self-supervised tokenizer with existing supervised tokenizers, such as FACodec, focusing on architectural design trade-offs and the effectiveness of our method in zero-shot timbre imitation tasks.
>
> We acknowledge that our proposed self-supervised tokenizer requires careful parameter tuning. However, it offers several notable advantages that the existing supervised tokenizers (like FACodec) do not have: (1) **simpler VQ design** with only one codebook and low bandwidth requirements (content/content-style tokens: 0.25/0.6 kbps), (2) **independence from annotated corpora**, and (3) **simpler and more stable training process**. These characteristics make it particularly suitable for scaling up in the current large-data era. While theoretically our current design may not be the optimal choice, our parameter configuration has already demonstrated promising results across various zero-shot voice imitation tasks. We plan to investigate the integration of self-supervised and supervised tokenizers in future work, aiming to potentially achieve both superior performance and enhanced interpretability (i.e., injecting prior domain knowledge from supervised learning).
>
> ```
> Style information leakage issue during acoustic modeling.
> ```
>
> Thank you for raising such an insightful point. Indeed, theoretically, both content-style tokens and unmasked mel spectrograms contain exploitable style information for reconstructing masked mel spectrograms. Specifically, unmasked mel spectrograms contain **global** information (such as the "consistent style information" you mentioned), while content-style tokens contain **frame-level** information (such as F0 and other prosodic information), which we consider to be **stronger style signals**. We observe that during reconstruction learning, models tend to "take the shortcut" and will preferentially fetch information from the input stream that contains stronger signals. Therefore, during the inference stage, the frame-level style information from content-style tokens should dominate. This is supported by our empirical results: compared to existing baselines, Vevo-Timbre demonstrates superior performance in metrics such as FPC and PS-MOS (i.e., prosody consistency to source).
>
> To better validate the impact of unmasked mel spectrograms (i.e., how to effectively inject timbre information), we conducted an additional comparative experiment. Specifically, we compared three approaches for incorporating timbre information:
>
> 1. **Vevo-Timbre (ours)**: Utilizes unmasked mel spectrograms, following the design of Voicebox [27], with in-context learning as its primary characteristic.
> 2. **Global speaker embedding**: For each sample, we extract global speaker embeddings using a [pre-trained speaker verification model](https://github.com/modelscope/3D-Speaker/tree/main/egs/3dspeaker/sv-cam++), similar to the approach adopted by CosyVoice [24].
> 3. **Global speaker encoder**: For each sample, we randomly select another sample from the same speaker as a reference. We use an ECAPA-TDNN-based speaker encoder to extract global timbre information from this reference. This approach is adopted similar to FireRedTTS [12].
>
> | Method                   | WER $\downarrow$ | Speaker SIM $\uparrow$ | FPC $\uparrow$ |
> | ------------------------ | ---------------- | ---------------------- | -------------- |
> | Vevo-Timbre (ours)       | 6.908            | 0.403                  | 0.797          |
> | Global speaker embedding | 7.077            | 0.385                  | 0.759          |
> | Global speaker encoder   | 8.121            | 0.369                  | 0.785          |
>
> As shown in the table above, Vevo-Timbre demonstrates superior performance across all metrics, especially in the Speaker SIM metric. We attribute these results to two key factors: (1) The content-style tokens we employ contain sufficiently rich style information, effectively mitigating potential information leakage from unmasked mel spectrograms. (2) Our current design effectively leverages the advantages of in-context learning, significantly enhancing the model's prompt imitation capabilities and overall performance.

---

> ### Author Response · Authors · 2024-11-19
> **Response by Authors (Part 2/2)**
>
> ```
> 100-hour training data for content and content-style tokenizers.
> ```
>
> Thank you for your insightful question. [In our common reply above](https://openreview.net/forum?id=anQDiQZhDP&noteId=OvBCMTxJ97), we provided a detailed analysis of how varying training data scales affect tokenizer effectiveness. Specifically, we found that when training the tokenizer on English-only audiobook data, a 100-hour dataset was sufficient to achieve an effective tokenizer, with minimal gains from further data scaling. We hypothesize this is due to the relatively simple distribution of such data. Furthermore, in our recent extension work, we discovered that for in-the-wild and multilingual data, which has a more diverse distribution, increasing the tokenizer's training data scale is quite beneficial. Detailed results can be also found in the common reply above.
>
> ```
> Evaluation beyond audiobook data.
> ```
>
> Yes, we have evaluated beyond audiobook data. Among our four evaluation sample sets, while LS originates from the audiobook domain, CV (CommonVoice), ACCENT, and EMOTION are all from non-audiobook domains (See Page 6, Line 319). In our recent extension work, we have applied Vevo to in-the-wild, multilingual scenarios. We plan to release comprehensive evaluation results across multiple evaluation settings in both the revised version of our paper and with our open-source release.
>
> ```
> How the "randomly mask ratio" determined.
> ```
>
> We follow the "audio model" parameter settings from Voicebox [27], including the masked mel spectrogram ratio, classifier-free guidance parameters, learning rate, etc. (See Appendix B.4).
>
> > [12] Fireredtts: A foundation text-to-speech framework for industry-level generative speech applications. arXiv 2024.
> >
> > [24] Cosyvoice: A scalable multilingual zero-shot text-to-speech synthesizer based on supervised semantic tokens. arXiv 2024.
> >
> > [27] Voicebox: Text-guided multilingual universal speech generation at scale. NeurIPS 2023.

---

> ### Author Response · Authors · 2024-11-25
> **Official Comment by Authors**
>
> Thanks again for your valuable comments, we would be grateful if we could hear your feedback regarding our answers. We would be happy to answer and discuss if you have further comments.

---

> ### Author Response · Authors · 2024-11-30
> **Looking forward to your feedback**
>
> Dear reviewer:
>
> Thanks again for the time and effort you have dedicated to reviewing our paper! We greatly appreciate your insightful comments and suggestions. We have put much effort into running new experiments to address your concern, and we would love to receive feedback from you. If your concern is addressed, we humbly invite the reviewer to consider increasing the score. Your support means a lot to us!

---

### Author Response · Authors · 2024-11-19
**Common Response to Two Key Issues (Part 1/2)**

We sincerely thank all four reviewers for the insightful comments and appreciation of our strengths including "simple yet efficient method" (Reviewer EEAE), "a meaningful attempt for disentangled representations" (Reviewer 1b8h), "impressive idea, results, and demo" (Reviewer z29c), and "the effectiveness of experimental results" (Reviewer EEAE, z29c, PTWX).

We would like to address two key issues raised by the reviewers in this common response. First, **the comparison between our self-supervised tokenizer and the existing supervised tokenizers**, as mentioned by Reviewers EEAE and 1b8h. Second, an exploratory experimental analysis concerning **the effect of different training data scales of our proposed tokenizer**, as noted by Reviewers EEAE and PTWX.

## Issue 1: Our Proposed Self-supervised Tokenizer vs. Supervised Tokenizer

**We have already performed comparisons between our proposed self-supervised tokenizer and several supervised tokenizers**, including **ASR tokens** (Table 2 and Table 4) and **FACodec** (Table 3). Specifically:

1. **ASR tokens**: They are extracted from [HuBERT-ASR-Large](https://pytorch.org/audio/0.10.0/pipelines.html#hubert-asr-large) (an encoder-only ASR model), which can be considered as a type of content tokens. As shown in Table 2, our proposed content tokens (i.e., VQ-VAE tokens, #Vocab = 32) achieve comparable intelligibility to ASR tokens (WER at 7.836 vs. 9.731). This is noteworthy as our method does not rely on explicit textual transcription for training. Furthermore, in zero-shot accent/emotion conversion tasks (Table 4), our proposed Vevo-Style (i.e., content tokens -> content-style tokens) has matched the performance of Vevo-Style (ASR) (i.e., ASR tokens -> content-style tokens). These results validate the effectiveness of our proposed self-supervised content tokens.

2. **FACodec**: In the upper sub-table of Table 3, we present a performance comparison between Vevo-Timbre (which is the "VQ-VAE tokens, #Vocab = 4096" in Table 2) and FACodec on the zero-shot timbre imitation task (i.e., content-style tokens -> acoustic representations, which is the same task as in Table 2). Our results show that Vevo-Timbre outperforms FACodec across all metrics including intelligibility (WER at 2.968 vs. 3.682), speaker similarity (S-SIM at 0.420 vs. 0.327; SS-MOS at 3.36 vs. 3.10), and style consistency (FPC at 0.686 vs. 0.611; PS-MOS at 3.45 vs. 3.10). These improvements demonstrate the effectiveness of our proposed self-supervised content-style tokens.

Additionally, we provide an in-depth analysis that outlines both the advantages and limitations of our proposed self-supervised tokenizer compared to existing supervised tokenizers, such as FACodec:

| Method                                 | Content Tokens                                 | Style Tokens                                    | Content-Style Tokens                                                                        | Strengths                                                                                  | Limitations                                                                                                                               |
| -------------------------------------- | ---------------------------------------------- | ----------------------------------------------- | ------------------------------------------------------------------------------------------- | ------------------------------------------------------------------------------------------ | ----------------------------------------------------------------------------------------------------------------------------------------- |
| **FACodec (supervised)**               | Two codebooks (K = 1024), 80Hz, 1.6 kbps  | Single codebook (K = 1024), 80Hz, 0.8 kbps | N/A. If combining content and style tokens: Three codebooks (K = 1024), 80Hz, 2.4 kbps | Leverages domain knowledge with high interpretability due to supervised training        | (1) Dependency on an annotated corpus. (2) Complex training involving multiple losses and adversarial elements (e.g., several GRL layers) |
| **Vevo's tokenizer (self-supervised)** | Single codebook (K = 32), 50Hz, 0.25 kbps | N/A                                             | Single codebook (K = 4096), 50Hz, 0.6 kbps                                             | (1) Independent of annotated corpus, highly scalable. (2) Simple and stable training process | (1) Codebook size requires careful design and tuning. (2) Lower interpretability compared to supervised tokens                             |

where K means the vocabulary size of the codebook.

**(... See Part 2 ...)** As shown in the comparison above...

---

> ### Author Response · Authors · 2024-11-19
> **Common Response to Two Key Issues (Part 2/2)**
>
> **(... Continued ...)**
>
> As shown in the comparison above, both supervised tokenizers and Vevo's self-supervised tokenizer present distinct advantages and limitations. We argue that our self-supervised approach offers three significant benefits: (1) **simpler VQ design** with only one codebook and low bandwidth requirements (content/content-style tokens: 0.25/0.6 kbps). (2) **independence from annotated corpora**, and (3) **simpler and more stable training process**. These characteristics are particularly valuable in the current era of large-scale data processing and merit further exploration by the research community. Notably, in Issue 2, we will demonstrate that our self-supervised tokenizer achieves further improvements in zero-shot voice imitation tasks when trained on larger-scale in-the-wild data. For future work, we plan to investigate the integration of self-supervised and supervised tokenizers to potentially achieve both superior performance and enhanced interpretability.
>
> ## Issue 2: Effect of Different Training Data Scales on Our Proposed Tokenizer
>
> We appreciate Reviewers EEAE and PTWX's questions regarding the training data scale of our proposed tokenizer. In our early experiments, we investigated the impact of different training data scales on tokenizer performance - specifically, exploring the effects of 10-hour, 100-hour, and 1K-hour audiobook data (English-only) on our proposed tokenizer. Additionally, in our recent extension work, we are exploring the effects of in-the-wild data (diverse acoustic environments; multilingual) on Vevo. Here we include some results from these experiments:
>
> | Training Data of Tokenizer | Training Data of Acoustic Model | WER $\downarrow$ | Speaker SIM $\uparrow$ | FPC $\uparrow$ |
> | -------------------------- | ------------------------------- | ---------------- | ---------------------- | -------------- |
> | 10 hours, Audiobook        | 6K hours, Audiobook data             | 7.032            | 0.387                  | 0.768          |
> | 100 hours, Audiobook       | 6K hours, Audiobook data             | 6.908            | 0.403                  | 0.797          |
> | 1K hours, Audiobook        | 6K hours, Audiobook data            | 6.759            | 0.396                  | 0.785          |
> | 1K hours, In-the-wild      | 10K hours, In-the-wild data          | 6.361            | 0.445                  | 0.796          |
> | 10K hours, In-the-wild     | 10K hours, In-the-wild data         | 5.861            | 0.525                  | 0.815          |
> | 10K hours, In-the-wild     | 100K hours, In-the-wild data       | 4.462            | 0.600                  | 0.806          |
>
> From the table above, we can draw two key observations:
>
> 1. **For English-only audiobook data, increasing the training data scale for the tokenizer yields only marginal improvements**. We hypothesize this is due to the relatively homogeneous distribution of English-only audiobook data, where a modest amount of training data suffices to achieve robust tokenizer performance.
>
> 2. **For multilingual and in-the-wild data, we observe more substantial improvements as the training data scale increases**. We attribute this to the more complex and diverse data distribution of such datasets, where larger training volumes contribute significantly to tokenizer's effectiveness and robustness. Furthermore, our results demonstrate that scaling up the acoustic model's training data in conjunction with the tokenizer leads to additional performance gains.

---

### Meta-Review · Area_Chair_Tjcn · 2024-12-08

**Metareview:**

The paper introduces Vevo, a zero-shot voice imitation framework with controllable timbre and style. It employs a fully self-supervised approach that progressively decouples speech's timbre, style, and linguistic content using VQ-VAE tokenizers combined with an information bottleneck technique. Vevo outperforms existing models in accent and emotion conversion, demonstrating particular excellence in zero-shot tasks.

The reviewers' opinions are diverse.

**Additional Comments On Reviewer Discussion:**

The paper initially received mixed feedback from four reviewers, with scores of 5, 5, 8, and 6. After the authors' first rebuttal, one reviewer (Reviewer 1b8h) changed from a borderline acceptance (5) to a rejection (3). The authors then submitted a second detailed response to Reviewer 1b8h, but the reviewer did not provide additional follow-up comments.


Reviewer EEAE and Reviewer 1b8h believe that the proposed approach requires meticulous parameter tuning and is a suboptimal choice. They suggest comparing it with supervised tokenizers like FACodec.

Authors' reply: The authors have conducted comparisons with several supervised tokenizers, including ASR tokens and FACodec. They also highlight several aspects where SSL approaches outperform supervised methods.


Reviewer EEAE raised concerns about style information leakage during acoustic modeling.

Authors' reply: To better assess the impact of unmasked mel spectrograms (i.e., the effectiveness of injecting timbre information), the authors conducted an additional comparative experiment. Specifically, they compared three approaches for incorporating timbre information. Among these, Vevo-Timbre demonstrated superior performance.

Reviewer 1b8h and Reviewer PTWX consider the approach lacking in novelty. Reviewer 1b8h also finds the approach similar to SeedTTS and CosyVoice. (However, CosyVoice can be disregarded as it is considered concurrent work based on the review guidelines of ICLR.)

Authors' reply: Vevo extends beyond being a zero-shot TTS system to serve as a unified and versatile voice imitation framework. In addition to zero-shot TTS, Vevo supports additional tasks such as zero-shot voice conversion and zero-shot accent/emotion conversion. The authors mentioned that the proposed content-style tokens effectively decouple timbre information, enabling timbre-controllable generation during the acoustic modeling stage. Furthermore, Vevo can process speech input in addition to text input. This feature enables functionalities like zero-shot accent and emotion conversion.

Note: It appears that Reviewer 1b8h remains unconvinced.

AC's note: The paper's primary focus is the disentanglement approach. Therefore, although the paper uses an architecture similar to previous TTS models, it does not detract from its novelty.


Reviewer 1b8h: The baselines are relatively weak.

Authors' reply: For zero-shot timbre and voice imitation tasks, they compared UniAudio and FACodec. For zero-shot style imitation tasks, they compared Conv-Speak and VoiceShop. For zero-shot TTS, they compared VoiceCraft, Voicebox, CosyVoice, MaskGCT.



AC's opinion: The paper's main contribution is its disentanglement approach. Therefore, even though the overall cascaded generation framework of Vevo is not highly innovative—since similar frameworks has been sued in SeedTTS and CosyVoice—its novelty can still be appreciated. The concern is that careful hyperparameter fine-tuning is required (as the authors also acknowledge in their reply). However, the authors have not provided further studies to demonstrate the robustness of the proposed approach in different settings (e.g., using different SSL models).

Overall, I am inclined to support acceptance due to the novel disentanglement approach. However, if the SAC decides to reject the paper, I am okay with that.

---

### Decision · Program_Chairs · 2025-01-22

Accept (Poster)